# Meta-analysis reveals that pollinator functional diversity and abundance enhance crop pollination and yield

B.A. Woodcock [1], M.P.D. Garratt[2], G.D. Powney [1], R.F. Shaw [3], J.L. Osborne [3], J. Soroka[4],
S.A.M. Lindström [5,6,7], D. Stanley[8], P. Ouvrard [9], M.E. Edwards[10], F. Jauker [11], M.E. McCracken[1],
Y. Zou [12], S.G. Potts[2], M. Rundlöf [6], J.A. Noriega [13], A. Greenop[1], H.G. Smith [6,14], R. Bommarco [15],
W. van der Werf [16], J.C. Stout [17], I. Steffan-Dewenter[18], L. Morandin[19], J.M. Bullock[1] & R.F. Pywell[1]

How insects promote crop pollination remains poorly understood in terms of the contribution of functional trait differences between species. We used meta-analyses to test for correlations between community abundance, species richness and functional trait metrics with oilseed rape yield, a globally important crop. While overall abundance is consistently important in predicting yield, functional divergence between species traits also showed a positive correlation. This result supports the complementarity hypothesis that pollination function is maintained by non-overlapping trait distributions. In artificially constructed communities (mesocosms), species richness is positively correlated with yield, although this effect is not seen under field conditions. As traits of the dominant species do not predict yield above that attributed to the effect of abundance alone, we find no evidence in support of the mass ratio hypothesis. Management practices increasing not just pollinator abundance, but also functional divergence, could benefit oilseed rape agriculture.

[1] NERC Centre for Ecology and Hydrology, Crowmarsh Gifford, Wallingford, Oxfordshire OX10 8BB, UK. [2] Centre for Agri-Environmental Research, School of Agriculture Policy and Development, University of Reading, Reading RG6 6AL, UK. [3] Environment & Sustainability Institute, University of Exeter, Penryn Campus, Penryn, Cornwall TR10 9FE, UK. [4] Saskatoon Research and Development Centre, Agriculture and Agri-Food Canada/Government of Canada, Saskatoon S7N 0X2, Canada. [5] Department of Ecology, Swedish University of Agricultural Sciences, 750 07 Uppsala, Sweden. [6] Department of Biology, Lund University, 223 62 Lund, Sweden. [7] Swedish Rural Economy and Agricultural Society, Kristianstad S-291 09, Sweden. [8] Botany and Plant Science, School of Natural Sciences, Ryan Institute, National University of Ireland, Galway H91 TK33, Ireland. [9] University Catholique do Louvain, ELIA, Croix du Sud 2/L7.05.14, 1348 Louvain-la-Neuve, Belgium. [10] Leaside, Carron Lane, Midhurst, West Sussex GU29 9LB, UK. [11] Department of Animal Ecology, Justus Liebig University, Heinrich-Buff-Ring, 26-32, 35932 Giessen, Germany. [12] Department of Environmental Science, Xi'an Jiaotong-Liverpool University, 215123 Suzhou, China. [13] Department of Biogeography and Global Change, National Museum of Natural Science, Madrid 28006, Spain. [14] Centre for Environmental and Climate Research, Lund University, Lund S-223 62, Sweden. [15] Department of Ecology, Swedish University of Agricultural Sciences, Uppsala SE-750 07, Sweden. [16] Centre for Crop Systems Analysis, Wageningen University, Wageningen 6700, The Netherlands. [17] School of Natural Sciences, Trinity College Dublin, Dublin 2, Ireland. [18] Department of Animal Ecology and Tropical Biology, Biocenter, University of Würzburg, Am Hubland, 97074 Würzburg, Germany. [19] Pollinator Partnership Canada, Head Office, 423 Washington Street, 5th floor, San Francisco, CA 94111, USA. Correspondence and requests for materials should be addressed to B.A.W. (email: bawood@ceh.ac.uk)

The role of insect pollination in enhancing crop yield and quality represents one of the most widely appreciated ecosystem services, not least for its contribution to the 580 million tons of oilseeds grown worldwide annually[1,2]. Outside of the importance of overall community abundance and species richness, the contribution of functional differences between species that facilitate pollination remain poorly understood[3–5]. The importance of functional differences can be seen in terms of the debate over the relative contribution of domesticated bees (e.g., honey bees) or wild pollinators in the delivery of pollination services[6,7]. These debates are underpinned by an acknowledgement that not all species are equally important for the pollination of a given crop. Indeed, there is evidence that economically significant pollination is the result of a relatively small number of species[8–10]. For this reason, pollinator community composition may influence the delivery of pollination services under different environmental conditions[11,12].

A detailed understanding of what aspects of community structure affect crop pollination is fundamental for the sustainable management of agricultural systems[4,9,11,13]. For example, the mechanisms by which pollinator communities affect yield may inform decisions about interventions targeted to benefit key pollinators. As single interactions between individual pollinators and a flower represent the underlying mechanism promoting intra-specific pollen transfer, summed visitation rates across species are often used as a proxy for pollination services, for e.g., see ref. [14–16]. However, species-specific pollen transfer rates mean that distinct pollinator communities, differing in both the species they contain and their relative abundances, may make very different contributions to yield[7,17,18]. Morphological and behavioural characteristics of pollinators that affect their capacity to provide pollination are typically referred to as effect traits. The distribution of these effect traits within a pollinator community is expected to have a pivotal role in pollination services[3,7,19,20]. However, this has often proved hard to empirically demonstrate.

There exist two principal hypotheses originating from the plant community literature that describe mechanism to define how functional differences between species can promote pollination. The first is the mass ratio hypothesis. This proposes that pollination success would be best predicted by the traits of the numerically dominant species[4,21,22]. Here, the traits of rare or infrequent species contribute little to the provision of ecosystem function, and as such functional diversity *per se* is less important than what traits are expressed by the species most likely to interact with a crop flower. Community weighted trait means have been used as metrics for quantifying dominant traits within a community[4,17] and have provided an approach for testing the mass ratio hypothesis[22]. The complementarity hypothesis, in contrast, predicts that communities with non-overlapping trait distributions would be more likely to promote pollination. For example, communities with diverse traits may be better able to provide consistent pollination under environmentally variable conditions[4,12,22,23]. Assessing complementarity has been achieved by quantifying the number of functionally similar species (effect groups) within a community[19,24,25]. Measures of functional diversity, such as functional divergence, also provide a continuous measure of complementarity[26].

In this study we link pollinator community structure to yield gains in oilseed rape (*Brassica napus* L.: Brassicaceae). This crop is grown in all continents except Antarctica and is one of the principal crops used in the production of edible oils and biodiesel[27]. Although partially wind pollinated, studies have identified positive effects of insect pollination on yield in oilseed rape, e.g.,[14,28–31]. Meta-analyses provide a statistical approach for integrating results from independent studies lacking consistent methodologies but testing a common hypothesis. Using this

approach, we test if differences between pollinator communities resulting from functional differences in morphology and behaviour explain variation in crop yield in addition to that explained by simple yield-abundance relationships. We test whether, and to what extent, (1) complementarity provided by non-overlapping effect trait distributions increased pollination[4,19,32]. We infer increased pollination from correlations between effect group richness or functional trait divergence with oilseed rape yield. We also test the extent to which (2) pollination is determined by the effect traits of the numerically dominant species, a test of the mass ratio hypothesis[4,7,21,22]. We infer this by testing for correlations between yield and community weighted trait means of the pollinators. We focus only on the correlative relationships between community structure and yield and do not consider other effects of pollination, such as its role in promoting crop quality including seed oil content[33]. We show in this paper that pollinator abundance is consistently important in predicting oilseed rape yield. However, functional divergence between species traits explained additional variance in the response of yield above that explained by abundance alone. This provides evidence in support of the complementarity hypothesis. For simplified artificial communities constructed within mesocosms there is also evidence that species richness is positively correlated with yield. Although community weighted mean values of several effects traits do show correlations with oilseed rape yield, taken individually within the meta-analyses these traits do not predict yield above that attributed to the effect of abundance alone.

## Results

**Description of the data sets.** We assess the impacts of insect pollinators from studies using artificial pollinator communities added to caged crop plants (mesocosms), as well as those assessing the effect on yield resulting from naturally occurring pollinator communities (field studies). Meta-analyses were undertaken separately on mesocosm and field studies. The field studies were predominantly from Europe, but some were from the USA and China. The data set used in the meta-analyses was based on seven mesocosm studies and 16 field studies (Tables 1, 2). From each study, we correlated oilseed rape yield and measures of pollinator community structure. We then assessed the relative strength and direction of these correlations for each meta-analysis. The 23 studies contained records from 20,591 individual pollinators (mesocosms $N = 1375$; field studies $N = 19,216$) and 57 taxonomic units. These taxonomic units included species level ($N = 36$) and genus level ($N = 19$) classifications, as well as functional groups (e.g. *calyptrate* flies and *Pieris* spp.). Under naturally occurring field conditions, the flies *Bibio marci* (Bibionidae) ($N = 6528$) and Calypterate spp. (predominantly *Delia* spp. (Anthomyiidae); $N = 3853$) were the most abundant, although the honey bee (*Apis mellifera*) ($N = 3848$) was the third most frequently recorded pollinator. Only seven species were used to create the artificial mesocosm communities with no individual study combining more than two species. For this reason only abundance and species richness metrics were derived for mesocosm studies.

**Abundance and species richness effects on yield.** Abundance of insect pollinators was used as a simple surrogate measure for the visitation frequency of pollinators to oilseed rape. For both mesocosm ($\mu = 0.58$, CI: 0.26, 0.79; $z = 3.25$, $z$-test: $P = 0.001$; excluding two outlier studies where Cook's distance > 1, see Supplementary Methods) and field studies ($\mu = 0.37$, CI: 0.24, 0.49; $z$-test: $z = 5.09$, $P < 0.001$) positive correlations were identified between yield and abundance (Figs. 1, 2). For mesocosm studies there was also a positive correlation between species richness and yield ($\mu = 0.62$, CI: 0.50, 0.72; $z$-test: $z = 7.85$, $P < 0.001$; excluding

### Table 1 Description of mesocosm-based studies

| Study | Country | N | Variety | Sterility | Taxonomy | Yield metric |
|---|---|---|---|---|---|---|
| M1: Jauker and Wolters[52] | Germany | 23 | Licosmos (Rest.Hyb.) | MF | Di. | Seeds silique$^{-1}$ |
| M2: Jauker et al.[28] | Germany | 28 | MSL 501C (Hybrid) | MS | Hy. Di. | Seeds silique$^{-1}$ |
| M3: Steffan-Dewenter[14] | Germany | 19 | Express MSL (Hybrid) | MS | Hy. | Seeds silique$^{-1}$ |
| M4: Steffan-Dewenter[14] | Germany | 19 | Express (Rest.Hyb.) | MF | Hy. | Seeds silique$^{-1}$ |
| M5: Garratt et al.[47] | UK | 70 | Heros (Conv.) | MF | Hy. Di. | Seeds silique$^{-1}$ |
| M6: Soroka, et al.[67]—1994 experiment | Canada | 10 | PC FU1981 (Hybrid) | MS | Hy. | Tonnes ha$^{-1}$ |
| M7: Soroka, et al.[67]—1995 experiment | Canada | 12 | PC FU1981 (Hybrid) | MS | Hy. | Tonnes ha$^{-1}$ |

These studies assess the impacts of abundance and species richness on oilseed rape yield under controlled experimental conditions. As the taxonomic breath of species in mesocosms is low (≤2) more complex community measures (e.g., functional divergence or CWM) were not assessed in these meta-analyses. Oilseed rape plants are either male sterile and male fertile (MS) or are all male fertile (MF). Studies are split by the variety of oilseed rape and year of observation. N = number of sample units defined as fields or mesocosms. Conv., conventional variety; Rest.Hyb., restored hybrid variety. The taxonomic range of the level of species identification includes hymenoptera (Hy.), Diptera (Fl.) and Lepidoptera (Bu.). Yield metric describes the units of the measure of yield. In all cases zero pollinator abundance mesocosm were used as controls

### Table 2 Description of field-based studies used in meta-analysis

| Study | Country | N | Variety | Taxonomy | Excl. cage | Yield metric |
|---|---|---|---|---|---|---|
| F1: Lindström et al.[31] | Sweden | 10 | Excalibur (Rest. Hyb.) | Hy. Di. | No | Tonnes ha$^{-1}$ |
| F2: Lindström et al.[31] | Sweden | 11 | Galileo (Conv.) | Hy. Di. | No | Tonnes ha$^{-1}$ |
| F3: Bommarco et al.[33] | Sweden | 20 | SW Stratos (Conv.) | Hy. Di. | Yes | g seed plant-1 |
| F4: Wessex—2013[a] | UK | 4 | DK Cabernet (Conventional) | Hy. Di. | Yes | Seeds plant$^{-1}$ |
| F5: Wessex—2013[a] | UK | 4 | PR46W21 (Rest. Hyb.) | Hy. Di. | Yes | Seeds plant$^{-1}$ |
| F6: Hillesden—2014[a] | UK | 12 | Excalibur (Rest. Hyb.) | Hy. Di. Le. | Yes | Seeds plant$^{-1}$ |
| F7: Salisbury—2012[a] | UK | 12 | DK Cabernet (Conv.) | Hy. Di. Le. | Yes | Tonnes ha$^{-1}$ |
| F8: Woodcock et al.[29] | UK | 4 | NK Molten (Conv.) | Hy. Di. Le. | Yes | Tonnes ha$^{-1}$ |
| F9: Woodcock et al.[29] | UK | 8 | DK Cabernet (Conv.) | Hy. Di. Le. | Yes | Tonnes ha$^{-1}$ |
| F10: Waddesdon—2013[a] | UK | 12 | Dimension (Rest. Hyb.) | Hy. Di. Le. | Yes | Tonnes ha$^{-1}$ |
| F11: Stanley et al.[54] | Ireland | 4 | Castile (Conv.) | Hy. Di. | Yes | Seeds silique$^{-1}$ |
| F12: Morandin and Winston[53] | USA | 16 | Advanta cv45A71 (Conv) | Hy. Di. | Yes | g seed plant$^{-1}$ |
| F13: Morandin and Winston[53] | USA | 20 | Advanta cvCL289 (Conv.) | Hy. Di. | Yes | g seed plant$^{-1}$ |
| F14: Morandin and Winston[53] 2002 expt. | USA | 32 | Monsanto cvDK3235 (Hyb) | Hy. Di. | Yes | g seed plant$^{-1}$ |
| F15: Morandin and Winston[53] 2003 expt. | USA | 19 | Monsanto cvDK3235 (Hyb) | Hy. Di. | Yes | g seed plant$^{-1}$ |
| F16: Zou et al.[48] | China | 34 | YangGuang-09 (Conv.) | Hy. Di. Le. | Yes | Seeds silique$^{-1}$ |

[a]Unpublished data set methodologies described in Supplementary Methods. Unpublished data provided in full in a Source Data file
These studies as used in the second meta-analysis are based on of observations of the impact of wild pollinator communities under typical agricultural conditions. In contrast to mesocosm studies it was possible to derive complex measures of community structure and functional divergence. Studies are split by the variety of oilseed rape and year of observation. Abbreviations are the same as those given for Table 1. All varieties assessed under field conditions are male fertile. The use of pollinator exclusion cages to directly assess impacts of seed set is indicated[68]

two studies where Cook's distance > 1), with this effect acting independently of abundance as a moderator (QM test of moderators: $QM_1 = 0.01$, $P > 0.05$). However, in field studies this correlation between species richness and oilseed rape yield was not found ($\mu = 0.05$, CI: −0.18, 0.28; z-test: $z = 0.42$, $P > 0.05$; excluding one study where Cook's distance > 1; Figs. 1, 2). Abundance did, however, act as a moderating effect of this relationship (QM test of moderators: $QM_1 = 20.1$, $P < 0.001$; $\mu = 0.77$, CI: 0.52, 0.90). There was no effect of either male sterility (mesocosm studies: QM test of moderators: $QM_1 = 0.1$, $P > 0.05$) or hybrid, restored hybrid or conventional breeding types (field studies: QM test of moderators: $QM_1 = 0.01$, $P > 0.05$) on the response between pollinator species richness and oilseed rape yield.

**Complementarity effects on oilseed yield**. We quantified the role played by complementarity in species traits by testing the relationship between functional divergence and effect group richness on oilseed rape yield. Due to the small number of species included in mesocosm studies (≤2) effects of functional community structure were only assessed for field studies. While functional divergence describes the extent to which trait distributions are non-overlapping, effect group richness counts the number of distinct clusters of pollinator species showing higher levels of within as opposed to between group similarities in effect traits. In support of the complementarity hypothesis, there was a positive correlation between functional divergence and yield. This was true when using either a scaled measure of functional divergence ($\mu = 0.47$, CI: 0.34, 0.58; z-test: $z = 6.25$, $P < 0.001$; excluding two studies where Cook's distance > 1; Figs. 1, 2) or an unscaled measure of functional divergence where control plots (pollinator exclusion cages without pollinators) had been excluded from the analysis ($\mu = 0.28$, CI: 0.01, 0.51; z-test: $z = 2.01$, $P = 0.05$; excluding three studies where Cook's distance > 1; Fig. 2). In both cases this effect was independent of abundance as a moderator of this relationship (QM test of moderators: scaled functional divergence: $QM_1 = 0.01$, $P > 0.05$; Functional divergence excluding control plots: $QM_1 = 0.09$, $P > 0.05$). Effect group richness was not correlated with oilseed rape yield ($\mu = 0.13$, CI: −0.14, 0.39; z-test: $z = 0.97$, $P > 0.05$; excluding one study where Cook's distance > 1), although this relationship was moderated by a significant positive effect of abundance (QM test of moderators: $QM_1 = 10.9$, $P = 0.001$; $\mu = 0.73$, CI: 0.39, 0.90). There was no

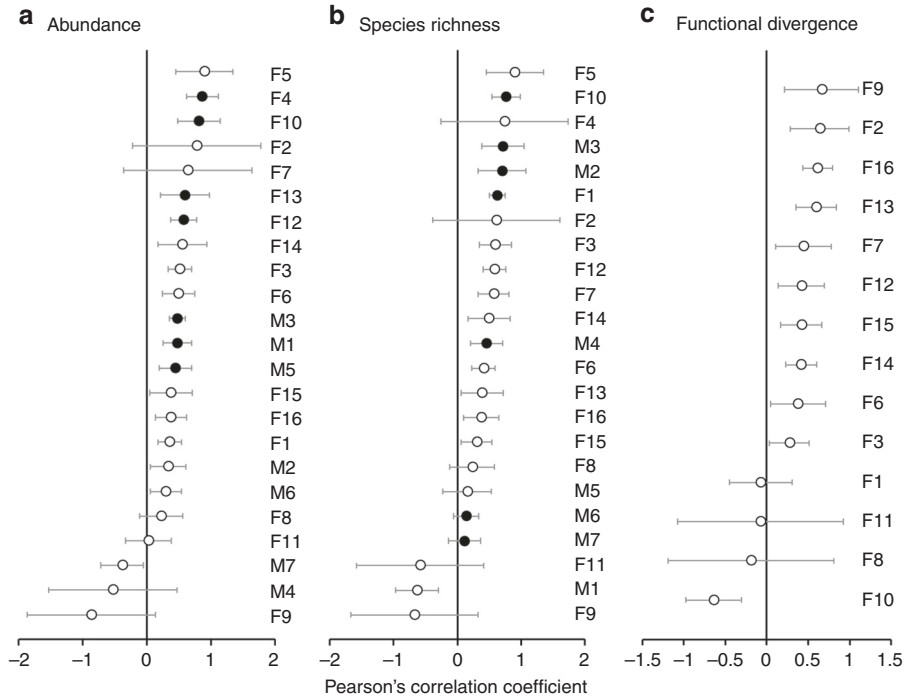

**Fig. 1** Correlations between oilseed rape yield and pollinator community structure. Pearson's correlation coefficient (**r**) for the **a** abundance, **b** species richness, and **c** scaled effect trait functional divergence of insect pollinators (error bars ±1 Standard Error) for individual studies. Studies originate from either naturally occurring pollinator communities observed under field conditions (open circle; $N = 16$) or artificial assemblages established in mesocosms (black circle; $N = 7$). Study abbreviations are given in Tables 1 and 2

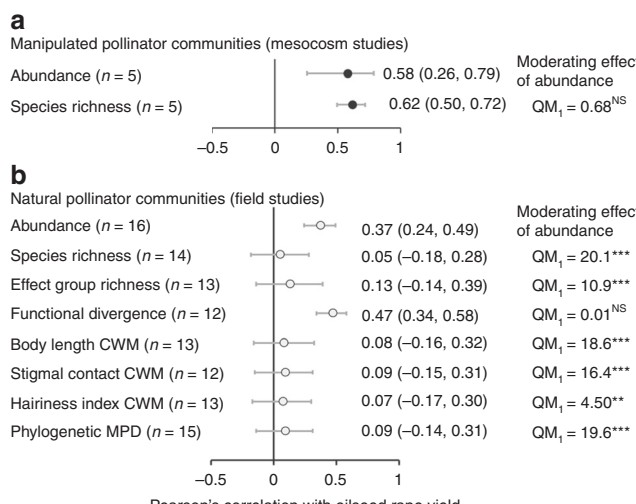

**Fig. 2** Oilseed rape yield and pollinator community structure forest plots. Mean correlation coefficient (**r**) for the relationship between oilseed rape yield and measures of pollinator community structure (error bars ± 95% credible intervals) for **a** mesocosm and **b** field-based studies. To test if the effect of pollinator community structure was responsible for changes in yield above that resulting from overall insect abundance the correlation between abundance and yield was included as a moderator in all models (P values relate to the QM test of moderators where this effect was tested). The exception for this was for models directly testing the effect of abundance. There was no significant effect of either male sterility (mesocosm studies) or varietal breeding type (hybrid, restored hybrid or conventional). Correlations are back transformed from Fishers z values and final sample size (n) follows removal of studies with high influence (Cook's distance > 1). Scaled functional divergence is shown, although results are qualitatively identical for measures when excluding pollinator control plots ($\mu = 0.28$, CI: 0.01, 0.51)

evidence that hybrid, restored hybrid or conventional breeding types acted as a moderator for the response of oilseed rape yield to either functional divergence (QM test of moderators: scaled functional divergence: $QM_1 = 0.51$, $P > 0.05$; functional divergence excluding control plots: $QM_1 = 3.37$, $P > 0.05$) or effect group richness (QM test of moderators: $QM_1 = 0.01$, $P > 0.05$).

Functional divergence is a composite measure derived from all 15 effect traits defined for the pollinators (Table 3). To provide insight into which of these traits may be contributing to the effect of functional divergence, we used general linear mixed models to test for correlations between oilseed rape yield (as a response) and linear combinations of the 15 effect traits described by their community weighted means (CWM) as explanatory variables. In contrast to the meta-analysis, this was based on individual plot values from the 16 field studies and did not attempt to partition out the relative contributions of CWM trait values from the effect of abundance alone. We assessed all model combinations excluding interactions (16,384 tested models) and from this derived a sub-set of 53 best fit models that fell within 2 AIC of the overall best fitting model ($\Delta AIC \leq 2$ sub-set). Models within this $\Delta AIC \leq 2$ sub-set had equivalent explanatory power for the data[34]. Five effect traits were represented in ≥25% of the models within the $\Delta AIC \leq 2$ sub-set (Supplementary Data 1). In all cases, these showed positive correlations between oilseed rape yield and the CWM trait values (Supplementary Figure 1). These traits were: (1) the presence of propodeal corbicula ($\Sigma w_i$ variable importance score = 0.58; model average correlation coefficient $\beta = 0.70$); (2) body length ($\Sigma w_i = 0.42$; $\beta = 0.02$); (3) the probability of stigmal contact when foraging ($\Sigma w_i = 0.37$; $\beta = 0.29$); (4) the hairiness index ($\Sigma w_i = 0.30$; $\beta = 0.27$); and (5) the presence of long tongues ($\Sigma w_i = 0.22$; $\beta = 0.13$). A summary model containing all five traits had a $R^2_{\mathrm{marginal}}$ of 0.17.

**Impact of dominant traits on oilseed yield**. To test the mass ratio hypothesis that the traits of the dominant species predict

**Table 3 Description of behavioural and morphological effect traits**

| Effect trait no. | Name | Description | Correlation (t-test) with pollen stigmal deposition |
|---|---|---|---|
| 1 | Body length | Body length is related to both inter tegular distance and body mass[69,70] and is inter-correlated with a wide range of functional characteristics[40], including foraging range in bees[41]. | $t_7 = 4.78^{**}$ ($r = 0.85$) |
| 2 | Mean time on flower | The mean amount of time (seconds) spent foraging on an oilseed rape floret. Data from Woodcock et al.[18] but augmented with unpublished data. | $t_7 = -1.13$ NS |
| 3–4 | Nectar or pollen foraging | The probability during a foraging event the pollinator will forage for nectar (trait 3) or pollen (trait 4). Data from Woodcock et al.[18], but augmented with unpublished data. | $t_7 = -0.71$ NS $t_7 = -1.31$ NS |
| 5 | Stigmal contact | The probability that stigmal contact will be made when foraging. Data from Woodcock et al.[18], but augmented with unpublished data. | $t_7 = 2.61^*$ ($r = 0.70$) |
| 6 | Dry pollen on body | The probability of presence of free dry pollen anywhere on the individual. Data from Woodcock et al.[18] but augmented with unpublished data. | $t_{11} = -1.31$ NS |
| 7 | Hairiness index | Hairiness affects pollen grain deposition on stigmas[37] and, in bees, is used to detect electromagnetic fields emitted by flowers as pollination cues[38]. For each species, body parts that contact oilseed stigmas (head, thorax, sternum, abdomen underside, femora, tibiae and meta-tarsus (legs assessed separately)) were scored as: (1) coarse setae or extremely short hairs; (2) short (c. basal tibiae 1 diameter) but dense hairs ($>50$ mm$^2$); (3) long ($>$basal tibiae 1 diameter) dense ($>50$ mm$^2$) hairs. This score was summed and given as a percentage of the maximum score of 24. | $t_7 = 2.44^*$ ($r = 0.67$) |
| 8 | Mouthpart type | The length of the tongue used to collect nectar affects host plant specialisation, and is defined as either long, medium or short[58]. A separate category is listed for insects with chewing mouthparts. | NA |
| 9–13 | Specific pollen collecting structures. | The presence of setae specifically used to collect pollen, listed by Michener[57] as the basitarsal scopa (trait 9), femoral corbicula (trait 10), strict tibial corbicula (trait 11), propodeal corbicula (trait 12) or abdominal corbicula (trait 13). Note these structures are associated with bees, however, their absence will affect the pollen carrying capacity and thus likelihood of pollen stigmal transfer of other species (e.g., for flies). | NA |
| 14 | Pollen carried only in the crop | Pollen carried only in the crop and, as such, not available for pollination[57]. As above, these structures are associated with bees, however, their absence may affect the likelihood of pollen contacting plant stigmas for other pollinating groups. | NA |
| 15 | Corbicula pollen moist | Pollen in corbicula storage structures may be either dry or moistened. Moistened pollen is less freely available for deposition onto plant stigmas[57]. As above, these structures are associated with bees, however, their absence may affect the likelihood of pollen contacting plant stigmas for other pollinating groups. | NA |

These were derived for each pollinator species or functional type of bee ($N = 44$), other Hymenoptera ($N = 1$), butterflies ($N = 1$) and flies ($N = 11$). To confirm the importance of these traits as predictors of pollination success (and so identify effect traits for assessment of the mass ratio hypothesis) they were correlated with a small sub-set of species where pollen stigmal deposition rates had been quantified[20,68] (Supplementary Methods). The significance of these correlations is shown. For some effect traits there was insufficient range in the trait characteristic to provide a correlation (indicated by NA)

pollination success, we tested for relationships between oilseed rape yield and community weighted mean trait values (CWM). We focused on a sub-set of three effects traits shown to be correlated with pollen stigmal deposition rates (Supplementary Methods). After accounting for the effect of abundance, there was no correlation with yield for either CWM body length ($\mu = 0.08$, CI: $-0.16$, $0.32$; $z$-test: $z = 0.66$, $P > 0.05$; excluding two studies where Cook's distance $> 1$), CWM probability of stigma contact ($\mu = 0.09$, CI: $-0.15$, $0.31$; $z$-test: $z = 72$, $P > 0.05$; excluding three studies where Cook's distance $> 1$) and CWM hairiness index ($\mu = 0.07$, CI: $-0.17$, $0.30$; $z$-test: $z = 0.56$, $P > 0.05$; excluding two studies where Cook's distance $> 1$) (Fig. 2). Abundance was a significant and positive moderator of the relationship for CWM body length (QM test of moderators: $QM_1 = 18.7$, $P < 0.001$; $\mu = 0.77$, CI: $0.51$, $0.90$), CWM probability of stigma contact (QM test of moderators: $QM_1 = 16.4$, $P < 0.001$; $\mu = 0.74$, CI: $0.45$, $0.89$) and CWM hairiness index (QM test of moderators: $QM_1 = 4.50$, $P < 0.001$; $\mu = 0.78$, CI: $0.53$, $0.91$). There was no evidence that hybrid, restored hybrid or conventional breeding types acted as a moderator for these relationships (QM test of moderators: CWM body length: $QM_1 = 0.08$, $P > 0.05$; CWM stigma contact: $QM_1 = 1.69$, $P > 0.05$; CWM hairiness: $QM_1 = 0.63$, $P > 0.05$). Note, it is

only when abundance is accounted for that the CWM trait values show no relationship with yield (Supplementary Figure 1).

**Phylogenetic mean pairwise distance (MPD).** Phylogenetic MPD was used to test if the response of yield was predicted simply by the level of phylogenetic complexity of the community. For the field-based studies, there was no evidence of a positive correlation between phylogenetic MPD and oilseed rape yield ($\mu = 0.09$, CI: $-0.14$, $0.31$; $z$-test: $z = 0.74$, $P > 0.05$; excluding one study where Cook's distance $> 1$; Fig. 2). While abundance was a significant moderator (QM test of moderators: $QM_1 = 19.6$, $P < 0.001$; $\mu = 0.77$, CI: $0.51$, $0.90$), this was not the case for the hybrid, restored hybrid or conventional breeding type moderator (QM test of moderators: $QM_1 = 0.01$, $P > 0.05$).

## Discussion

Theses meta-analyses found evidence in support of the complementarity hypothesis that predicts that communities with non-overlapping trait distributions would be more likely to promote pollination. This was inferred from correlations between functional divergence in effect traits and oilseed rape yield that were

found after considering the effects of overall pollinator abundance. This emphasises that not all individuals are functionally equivalent and that species specific differences in effect traits can act to modulate how insects in a community can deliver pollination services[4]. As these relationships between pollinator community structure and yield were correlative, this does not represent a direct experimental demonstration of the complementarity hypothesis. However, the use of the meta-analysis approach to integrate findings from multiple studies does provide important evidence that the magnitude of functional differences between species plays a contributory role in predicting the yield of oilseed rape.

Correlations between species richness and yield suggest functional differences between species contribute to pollination, but do so under the assumption that species are equally distinct, independent of actual inter-specific functional differences. Correlations between species richness and yield were only found for the limited number of mesocosm studies that were assessed, with no significant relationship being identified for field-based studies where naturally occurring communities pollinated oilseed rape. The low species richness of mesocosm studies (≤2) may explain why only studies that use this experimental design identified an effect of species richness on yield. Mesocosms studies were also composed of similar species (i.e., those suitable for captive rearing) and as such the response of yield may represent special cases resulting from a sub-set of species interactions not necessarily generalisable to those of more complex communities. However, as mesocosm experiments are often designed to control for confounding factors, including abundance, they do provide useful mechanistic insights into the importance of species richness. Such effects may be harder to detect under field conditions, not least because community structure is estimated by sampling in such experiments and so exact measures of species richness are not known.

Moving beyond simple species richness, the number of clusters of species interacting with the crop in biologically similar ways provides an indication of how many functionally distinct groups of species are pollinating a crop. As such, there is no longer an assumption that species are all equally functionally distinct, but rather allows for some species being more or less similar to others. Through the proposed mechanism of complementarity, the number of functional groups of species in a community (defined in this study as effect groups) has been correlated with a variety of ecosystem functions[19,24,25,35]. This includes the yields of pumpkins resulting from insect pollination[19]. However, unlike the obligate cross-pollinated pumpkins, this meta-analysis failed to identify a correlation between effect group richness and yield for the predominantly wind pollinated oilseed rape (above that predicted by pollinator abundance alone). While the absence of a correlation with effect group richness did not support the complementarity hypotheses, this was not the case when complementarity was assessed using functional divergence in effect traits. This index provided a continuous and thus more biologically realistic measure of the extent to which trait-distributions were non-overlapping[26]. The correlation between functional divergence and oilseed rape yield supported the complementarity hypothesis as a mechanism describing how pollination is enhanced by insect communities[36].

As functional divergence is a composite index derived from many effect traits, a subsidiary analysis was used to identify, which sub-set of these effect traits played an important role in defining the link between insects communities and the pollination services they provide. Of these traits, the probability of stigmal contact represents a key limiting factor to pollination that describes the likelihood of contact between a pollinators body and the reproductive part of the plant, likely to be a prerequisite for pollen transfer[18]. The extent to which pollinator bodies are covered with fine hairs may also interact with stigmal contact by increasing the surface area over which pollen grains can stick and thus be transferred when stigmal contact is made[37]. The degree to which pollinator bodies are covered by hairs may also play a less obvious role in pollination. Mechanosensory hairs are used by some bees to detect electromagnetic fields provided by flowers as pollination cues[38]. Pollinators able to detect such cues may be more likely to achieve pollination, particularly where those cues are used by plants to identify flowers that have reached maturation. However, at present there is no direct evidence that this may be occurring in oilseed rape. Other important effect traits were associated with specific bee genera known to be common pollinators of oilseed rape, particularly in Europe. Specifically the propodeal corbicula associated with members of the genus *Andrena*, as well as the long tongues and large body sizes associated with *Bombus*[36]. It is quite likely that these effect traits may act as surrogates for clusters of other unmeasured but inter-correlated effect traits that also contribute to the importance of these bee genera for oilseed rape pollination.

The importance of effects traits linked to specific bee genera also emphasises a phylogenetic component to functional diversity, where common evolutionary history results in similar functional characteristics of species[39]. Identifying an underpinning and independent influence of insect phylogeny on pollination may be more pertinent to the study of evolution as opposed to crop management. Ultimately, pollinator communities are often phylogenetically constrained, not least due to their dominance by closely related bee species. However, while functional divergence did predicted yield, this was not the case for the considered measure of phylogenetic community diversity. It seems likely that while functional differences in both response and effect traits between species would be expected as a result of divergent phylogenetic histories[39], it is the complementary role of specific effect traits that impact on pollination success[40,41]. Such effect traits may be less predictable by phylogenetic community structure.

Once the overall yield-abundance relationship was accounted for, the effect traits (body length, stigmal contact behaviour and the hairiness index) of dominant species did not correlate positively with oilseed rape yield. As such, we found no direct evidence in support of the mass ratio hypothesis[4,21,22]. It is perhaps not surprising that once the effect of overall abundance was accounted for that the importance of CWM (being derived in part from measures of pollinator abundance) as a predictor of oilseed rape yield would disappear. Individual species abundances are a product of, among other things, complex competitive interactions and responses to local environmental conditions. As such, community weighted means represent a relatively simplistic way of assessing the mass ratio hypothesis. Indeed, as described above, there was evidence that aggregates, rather than individual effects traits, could affect yield when considered in combination. The limited data available for assessing the importance of individual traits was also a potential problem in the current approach. Ultimately only a small number of effect traits were used to derive CWM, with these based on correlations with published stigmal deposition rates used to validate the importance of a particular trait in the provision of pollination services[20,42] (Table 3; Supplementary Methods). As these stigmal deposition rates were available for only a few species, other effects traits (either identified or not identified in this analysis) may have been more relevant for predicting pollinator success in oilseed rape.

In conclusion, this meta-analysis provided evidence that, in addition to the underlying importance of overall visitation rates (described by the abundance proxy), complementarity between functionally distinct species was crucial to maximising yield potential for oilseed rape[4,19,21,22]. While this hypothesis was

supported by a correlation between functional divergence in effect traits and yield, these findings are based on a data set biased to the Northern Hemisphere and Europe in particular. As such, these results may not necessarily be generalisable to other regions, particularly if those regions are characterised by functionally different pollinator communities interacting with the same crop. However, the effect of complementarity on yield, as predicted by community functional divergence, may be expected to have relevance to other regions where oilseed rape is grown. Even where a fauna is taxonomically distinct to that considered here these novel communities would likely still show similar levels of variation in the effects traits we have considered. An increase in functional divergence as explained by these effect traits could similarly be expected to have a positive impact on oilseed rape yield.

Management practices that not only increases the overall abundance of pollinators (e.g., by placing the honeybee hives adjacent oilseed rape fields), but also increase the functional divergence of the overall community, could represent a practical approach to increasing yields in combination with conventional agronomic practices[4,9,13,43]. Such management tactics may include the targeted creation of specific breeding sites, for example bare ground to provide breeding sites for ground nesting bees, like *Andrena* spp.[18,44]. Similarly, field margins could be established with plants that support specific feeding associations or, through their flower structures, key foraging resources for certain species[44]. For example, long of short corolla flowers could be used to promote shorter (e.g., hoverflies) or longer tongued pollinator species (e.g., some *Bombus* spp.) respectively. Finally, as aspects of landscape structural complexity, like the availability of semi-natural habitats, can directly affect functional diversity, its manipulation may also be used as a management tool for enhancing functional complementarity[45,46]. Targeted management with the sole goal of enhancing the representation of certain species with key effects traits was, however, found to be unlikely to promote pollination. Even if evidence was found in support of this, management to enhance individual species with a particular trait may only represent a short-term solution to maximising pollination. Such approaches ignore the resilience provided by communities that have high diversity in other aspects of community structure, for example response traits[11,12]. Furthermore, any management that aims to increase the abundance of a limited number of functionally important species may have wider detrimental effects, particularly where those species are not of equal importance for phenologically different crop types grown elsewhere in the landscape[47]. Ultimately, from the perspective of maintaining profitable farming systems, management decisions will ultimately be dictated by the cost of interventions in relation to the expected increase in yield linked to the promotion of pollination. While this study focuses on oilseed rape, it has important implications for the role of insect pollination in general. However, crops with different breeding types or morphologically distinct flowers may have different dependencies on insect pollinators, with distinct effect traits to those considered here potentially having greater significance in terms of their impacts on yield. Further research is required to refine these relationships to maximise the potential for targeted management to support agricultural production. Independent of this factor the potential for even small contributions to yield or crop quality, resulting from management aimed at maximising the functional divergence of the species within a community, may mean the difference between profit and loss in high value crops[31,33,48].

## Methods

**Study criteria**. The process of identifying studies for inclusion in the meta-analyses is outlined in a Preferred Reporting Items for Systematic Reviews and Meta-

Analyses (PRISMA) flow diagram in Supplementary Methods. In summary, a Web of Science search under the criteria Oilseed rape OR Canola OR Rapeseed OR Brassica napus AND Pollination/Pollinator(s) AND Yield was undertaken. This was complimented by additional experiments sought from other sources, including published and unpublished studies (see Supplementary Methods for methodologies used to derive data in these unpublished studies where included in the meta-analyses). This produced a total of 145 experiments. These were checked for eligibility on the basis of: (1) they contained a direct measure of oilseed rape yield recorded and associated within individual experimental units; (2) insect pollinator communities were quantified to species or similar high-resolution taxonomic units (see below for details); (3) studies contained at least four experimental units allowing a measure of variance to be derived. This resulted in a sub-set of 18 experiments although these often contained observations on more than one variety of oilseed rape (Tables 1, 2). These experiments show bias to both the Northern Hemisphere and specifically to Europe. The experiments were based on two distinct methodologies. The first methodology was represented by mesocosm experiments, where a defined pollinator community was added to caged oilseed rape plants with the goal of assessing the effect on yield (Table 1). The second type of experimental design was based on field observations, where oilseed rape was grown under normal agronomic conditions and visited by a naturally occurring pollinator community composed of both wild and domesticated pollinators (Table 2). In all subsequent analyses experiments from these two distinct methodologies were analysed separately. For field studies, we did not consider landscape factors. While landscape setting can be a key predictor of pollinator community structure, we are directly focused on what communities are present and interacting with the crop at a site, rather than from where the pollinators originated.

Throughout the experiments, a range of varieties of oilseed rape were investigated (Tables 1, 2). The dependency of the oilseed rape on biotic pollination and the attractiveness of the crop to pollinators are affected by the variety and whether that variety is the product of a conventional open pollinated or hybrid breeding system[49,50]. Each experiment was sub-divided into studies that included observations undertaken on only a single variety within a single year. Under this criterion, studies with fewer than four replicates (field plots or mesocosm cages) were excluded as variance measures for the meta-analysis could not be derived[51]. After sub-setting the meta-analyses were based on seven mesocosm studies (based on 181 experimental units) and 16 field studies (based on 222 experimental units). The field studies were undertaken predominantly in Europe ($N = 11$; UK, Ireland and Sweden), although studies from the USA ($N = 4$) and China ($N = 1$) were also included (Table 2).

As zero abundance controls represent standard methodologies for assessing the contribution of pollinator communities to increasing crop yields, these were included in all analyses where available[8,28,29,33,47,52–54]. Zero abundance controls were found in all but two of the 23 studies; both exceptions occurred under field conditions (Table 2). For mesocosm experiments, individual mesocosms were treated as replicates. For field experiments, a plot observed for a single year was treated as a replicate, and data on pollinator communities assessed at shorter time scales were summed. Field experiments that included zero abundance controls in the form of exclusion cages were treated as separate data points equivalent to those used for the mesocosm studies.

**Oilseed rape yield metrics**. Yield was always based on the average recorded value for either an individual field or mesocosm cage. There was no common measure of yield, instead we used the most frequently derived metrics across all studies: seeds per silique (seed pod), seeds per plant, total seed weight per plant, and tonnes ha⁻¹ (Tables 1, 2). Subsequent meta-analysis included yield metric as a random effect to account for these study differences. Replicates from each experiment were standardised to have a mean of zero and standard deviation of one[55].

**Pollinator communities**. Individual field studies identified Hymenoptera (e.g., Apoidea, Vespidae, and Tenthredinidae), Diptera (principally Syrphidae), and Lepidoptera to a species, genus or functional level (Supplementary Data 2). Genus classifications were used when reliable identification to species was not consistent across studies (e.g., *Lasioglossum*, *Hylaeus*, *Megachile*, *Halictus* and *Osmia*). Functional types were used for taxonomically complex groups. For example, the predominately Calypterate flies, while composed principally of *Delia* spp. (Anthomyiidae), included other families such as Calliphoridae and Muscidae. Similarly, as 97.5% (total $N = 1139$) of all butterfly individuals were *Pieris* spp. (Pieridae), other butterfly species were combined together with them into a single functional group. For mesocosm studies, exact species compositions and abundances were always known. However, for field-based experiments abundance was quantified at the scale of a field using either pan traps or transect/quadrat-based observations. Species with ≤5 individuals across all 23 study data sets were excluded to minimise the effect of potentially transient species moving though fields but not foraging directly on the crop. Although coleopteran and parasitic Hymenoptera were recorded in the case of some studies, the variable taxonomic resolution meant that these were excluded from the analysis. Within each study abundance values were transformed to have a common standard deviation of one, although were not corrected to a mean of zero so that zero abundance plots remained zero. For each experimental replicate (either field or mesocosm) a summed abundance and species richness was derived.

**Effect traits**. We derived behavioural and morphological traits that had a high likelihood of affecting the success and rate with which pollen is transferred to the stigmas of oilseed rape (Table 2; Supplementary Data 2, 3). These are referred to here as effect traits. Traits were chosen that could be derived for a large number of species to assess whole community functional effects on oilseed rape pollination. Where possible traits were derived at the species taxonomic level, however, generic or functional group aggregates were used with mean trait values based on those individuals identified to species in at least one of the studies. Although representing a compromise approach dictated by taxonomic resolution, this allows the derivation of complex effect trait values at community scales at a biologically meaningful resolution. Fifteen effect traits were derived falling into the following categories: Trait 1) body length, which is related to both body size and in the case of bees inter-tegular distance[41,56]; Traits 2–6) quantification of behavioural interactions with oilseed rape flowers (e.g., time spent on flowers, pollen foraging and dry pollen on bodies)[18]; Trait 7) an index of overall body hairiness (see Table 2 for description), reflecting evidence that hair density affects pollen grain stigmal deposition[37]; Traits 8–14) morphological characteristics affecting pollen retention on bodies linked to the presence of corbicula and scopa[57]; Trait 14–15) pollen availability dictated by whether or not pollen is carried within bee crops[57]; Trait 15) mouthpart structure, classifying pollinators as having short, medium or long tongues[58], with a further category for insects with chewing mouthparts. Note, traits 8–15 are associated with bees, however, their absence will affect the pollen carrying capacity for non-bee species and as such are relevant cross taxon effect traits.

**Effect group richness**. While individual pollinator species are defined by unique sets of effect traits, broad similarities exist within certain clusters of species[35]. Such clusters (referred to as effect groups) are characterised by species with a higher level of within group similarity than is seen among other species in the community. The number of effect groups in a community provides an indication of the spread of the niche space within these communities. This provides a measure of complementarity by describing the extent to which trait distributions are non-overlapping[4,19,32]. To define the effect groups, we used Ward's method to hierarchically clustered species based on the matrix of the 15 effect traits described above[35]. Multi-scale bootstrap resampling was then used to calculate approximate unbiased (au) P values for each split of the hierarchy. Species were then aggregated into functional groups using $\alpha = 0.95$ as a threshold within the pvclust package in R V3.5.0. This approach defined five effect groups, with a further three species not allocated to any cluster. These were grouped to form a sixth effect group (Supplementary Methods). Effect group richness was defined as the number of effect groups represented in each experimental plot (e.g., field).

**Functional divergence**. The complementarity hypothesis assumes that communities with non-overlapping trait distributions will be more likely to promote increased pollination[4,19,32]. Functional divergence describes the extent to which species are either clumped or spread out in trait space[26] and as such represents a relevant metric for assessing complementarity. Other common diversity indexes, such as Rao's, measure different aspects of functional diversity. Functional divergence is low when most individuals in a community have traits near the centre of functional trait space and is greatest when individuals are positioned at the edges of the trait space. The functional divergence metric FDiv was derived for each experimental unit, although only for field-based studies. Functional divergence was calculated from a species presence-absence matrix to minimise the extent to which individual species abundance affected this metric. Functional divergence was derived in the FD package implemented R 3.5.0[59,60]. For studies that included control plots, FDiv was quantitatively similar to a binary covariate describing plots with or without bees (Supplementary Figure 1). We applied two separate approaches to address this issue, the first being to rescale our measure of functional diversity while retaining a comparison with control plots lacking pollinators. For each study in which FDiv values were greater than zero, values were corrected to FDiv minus the lowest non-zero FDiv value for that study. The second approach was to derive correlations between yield and FDiv after having excluded all control plots.

**Community weighted means (CWM)**. CWM represent abundance weighted trait values averaged across a community. They have been widely used to provide a simple measure of how dominant a trait is in a community and as such have been used to provide evidence for the mass ratio hypothesis[4,21,22]. While we use CWM to assess the trait values of the dominant species, these are defined for single traits at a time and as such they overlook trait variation among species within communities. We restricted our analysis of CWM to a sub-set of traits that can be demonstrated to be directly correlated with intra-specific pollen transfer. To do this we identified the presence of correlations between our derived traits and those taxonomic units in our data set where stigmal pollen deposition rates were available from published data, albeit from the close con-generic relative of oilseed rape, *Brassica napus*[20,42] (Table 3; Supplementary Methods). Where Pearson's correlations were identified between individual traits and stigmal pollen deposition rates, we derived CWM for field-based studies. In a number of cases it was not possible to assess correlations as there was insufficient trait variation for those nine species in our data set where published pollen stigmal deposition rates were available. CWM

trait values were derived for body length, body hairiness index, and the probability of stigmal contact behaviour.

**Phylogenetic mean pairwise distance (MPD)**. As phylogenetically distinct species also tend to be functionally distinct, there is potentially an underlying link between trait diversity and phylogeny. Indeed measures of phylogenetic diversity have been proposed as surrogate measures of functional diversity[39]. However, it is likely that measures of phylogenetic diversity predict the breath of all functional characteristics of a species (both response and effect traits), and so are not necessarily relevant to the effects trait approach considered here. We derived the Phylogenetic MPD to test if a response by oilseed rape yield was the result of phylogenetic differences in the communities, rather than a more specific measure of effect trait composition (either CWM trait values of functional divergence). MPD was derived using the Picante[61] package in the R 3.5.0 statistical environment based on a phylogeny derived from the species taxonomic associations (Supplementary Data 2). Phylogenetic distance was based on Grafen branch lengths.

**Statistical analysis**. We used a mixed effects meta-analysis to test the null hypothesis that oilseed rape yield showed no response to any measure of pollinator community structure (e.g., abundance, species richness, effect group richness, functional divergence and CWMs). Mixed effects meta-analysis treat correlations between pollinator community structure and yield from individual studies as random samples taken from a theoretical population and use these to produce summary correlation coefficients for that overall population. Each meta-analysis was based on Pearson's correlation coefficient ($r$) between yield and the measure of community structure transformed using Fisher's z with a variance of $1/(N-3)$ ($N$ = study replicates)[51]. In some field studies there was no effective variation between plots in certain community metrics (e.g., functional divergence). For these sites Pearsons correlation coefficients could not be derived. Separate meta-analyses were undertaken for data originating from mesocosm ($N = 7$) and field ($N = 16$) studies. Due to the small number of species found in mesocosms ($\leq 2$) measures of functional community structure (functional divergence, effect group richness and CWMs) were neither derived nor tested. As metrics of community structure are typically affected by the overall abundance of individuals in the community (a proxy for visitation rate to flowers) we tested whether functional metrics of pollinator community structure increased yield over and above that resulting from the effect of abundance alone. To do this we included the correlation coefficient (Fishers-z transformed) for the relationship between abundance and yield as a moderator in all models, except those directly testing the effect of abundance as a main effect. For mesocosm studies oilseed rape male sterility was included as a moderator as this has previously been shown to affect the importance of insect pollination[14]. In the case of field studies all crops were male fertile. However, conventional open-pollinated (the product of classic line-breeding methods or hybrid restored lines) as well as male sterile hybrid varieties (grown from male sterile and fertile parent lines) were grown[50]. As such breeding type (hybrid, restored hybrid or conventional) was included as a moderator for these analyses. The inclusion of breeding type in the analysis of mesocosm studies was not possible as it co-varied with male sterility. Yield metric was included as a random factor to account for between study differences in the way this was recorded. For all meta-analyses standard influence diagnostic plots were run and assessments of publication bias were undertaken using funnel plots[62]. To ensure robustness, studies showing high levels of influence (Cook's distance > 1) on estimates of correlation coefficients were excluded (see Supplementary Methods)[62,63]. Omnibus (QM) tests of individual moderators were undertaken and used as a basis for model simplification. We derive z-values for individual estimates of the correlation coefficient and use 95% credible intervals (CI) to confirm these[62]. Meta-analyses were performed in the R 3.5.0 statistical environment using the Metafor package[62].

While the meta-analysis focuses on how individual aspects of pollinator community structure affect yield, it is mechanistically important to understand if specific combinations of effects traits play an important role in promoting yield. To do this we used the general linear mixed model approach[64] to identify specific combinations of effects traits that are correlated with oilseed rape yield. Using individual plot level data for the field based studies only ($N = 222$ from 16 studies), we correlated average plot yield (corrected to have a SD of 1) with community weighted means of each of the derived effect traits (see Table 3). General linear models were implemented using the lme4[65] in R 5.0 and included as random effects study nested within yield metric nested within breeding type (hybrid, restored hybrid, or conventional). Note, that where effects traits were composed of nominal categories (long, short, medium tongue, and chewing mouthparts) these were treated individually as binary and a separate CWM was calculated for each level of the trait. As CWM values for chewing mouthparts, basitarsal scopa and pollen carried in the crop were data poor (10% of values were >0) these were excluded from the analysis. Due to high covariance between the CWM values of moist corbicula pollen and CWM strict tibial corbicula ($r = 0.98$) only the latter was included as a covariate. Rather than trying to define a single best fit model, we applied an information theoretic approach[34] and assess all potential model combinations, excluding interactions (16,383 models based on 14 explanatory variables). Individual model fit is described using Akaike's Information Criterion (AIC). AIC represents a measure of model fit that is weighted by the number of parameters in the model. Models falling within 2 AIC points of the best fit model

(referred to as a ΔAIC ≤ 2 sub-set) have broadly equivalent explanatory power for predicting the response of yield[34]. For this sub-set of models Akaike weights ($w_i$) were derived. These describe the probability that a given model would be selected as the best fitting model should the data be recollected under identical conditions. The importance of individual CWM fixed effects within the ΔAIC ≤ 2 sub-set was then assessed by summing the $w_i$ values of all models containing that explanatory variable (Σ$w_i$). This represents a variable importance parameter which ranges between 0 and 1, the higher the value the more important the explanatory factor. We focus only on those fixed effect CWM trait values that appear in at least 25% of the models found within the ΔAIC ≤ 2 sub-set. These have the greatest evidence for predicting oilseed rape yield. Average model parameter estimates weighted by their Akaike weight were derived from the ΔAIC ≤ 2 sub-set[34]. This analysis was undertaken using the MuMIn package[66].

**Reporting summary**. Further information on experimental design is available in the Nature Research Reporting Summary linked to this article.

## Data availability
Species abundance data from the 17 sites (including unpublished data sets) is given in Supplementary Data 4. The Source Data file contains all the data used in this paper including raw and corrected abundance data and derived plot level community metrics used to determine Pearson's correlations on which the meta-analyses were based. This includes data from previously unpublished studies. This file also contains all values presented in the figures. Supplementary Data 2 & 3 contains all trait data used in the derivation of functional divergence and CWM values.

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

## Acknowledgements

This study was funded by the Natural Environment Research Council (NERC) under research programme NE/N018125/1 ASSIST – Achieving Sustainable Agricultural Systems www.assist.ceh.ac.uk. ASSIST is an initiative jointly supported by NERC and the Biotechnology and Biological Sciences Research Council (BBSRC). Additional funding for field studies was from the Wessex Biodiversity Ecosystem Services Sustainability (NE/J014680/1) project within the NERC BESS programme. Other datasets were generated from research funded by: (a) the Insect Pollinators Initiative programme funded by BBSRC, Defra, NERC, the Scottish Government and the Wellcome Trust, under the Living with Environmental Change Partnership; (b) Defra project BD5005: Provision of Ecosystem services in the ES scheme; and (c) Irish Government under the National Development Plan 2007–2013 administered by the Irish EPA. Thanks to Matt Heard.

## Author contributions

B.A.W. and R.F.P. conceived the study. B.A.W. undertook all analyses with help from J.N., A.G. and G.P. and wrote the paper with significant contributions from all co-authors. M.G., R.S., J.O., J.St., I.S.-D., A.G., S.L., D.S., J.So., P.O., F.J., J.B., M.M., Y.Z., S.P., M.R., H.S., R.B., W.W., L.M. and S.D. contributed data sets. M.E. derived behavioural trait metrics.

## Additional information

**Competing interests:** The authors declare no competing interests.

