## [Peer Review File · Nature Communications]

Reviewers' Comments:

Reviewer #1:

Remarks to the Author:

This paper presents an analysis of several studies of the statistical relationship between various measures of the flower visitor community structure and yield, for oilseed rape. Whilst it is a very useful and interesting approach, and provides original insight, there are some issues with how the paper is written. The theoretical background is not well explained, key details are missing from the methods and there are several careless typing errors.

For it to be publishable in Nature Communications, it would need to be substantially revised to improve the quality of reporting and argument. Also, I have a concern about how the studies were identified. There may be other studies of oilseed rape pollination in the literature that are not included here, and the paper does not give me any confidence that the authors searched for them.

I am uneasy that there is no critical discussion about whether relationships between flower visitor community structure and yield actually reflect pollination services. An unspecified number of the 16 field studies included exclusion cages (lines 280-282), and therefore were measuring a service provided by pollinators. But this is not true for all the studies. This really needs to be clarified.

Detailed comments:

Line 69 : Change to 'management practices that increase pollinator abundance'

Line 96: introduction of the term 'effect traits' – I am not familiar with this term, so it would be good to have a reference here that defines it, or better still, a definition.

Lines 111-122: I don't find this a very clear description of the research question, approach and hypotheses. As this is a meta-analysis, you should at least state here what type of studies you aimed to include, from what sources you searched for them. Then develop research questions and hypotheses to test with the resulting dataset. The text sort of does this, but it's very vague. For instance: 'using a meta-analysis approach'. What does this mean?

There is very little explanation of the competing hypotheses being tested (complementary and mass ratio). Please expand on these, and explain them properly. For example, your results section implies that the complementarity hypothesis predicts a positive correlation between functional divergence and yield, but you haven't really explained this in the introduction. There is a statement of this in the methods (line 347-349) but without references.

The mass ratio hypothesis seems to come from plant ecology (reference 29-31). It seems oddly named and I miss an explanation of what it is, and why it might apply in this context.

Lines 119-120, you claim to test "to what extent, (1) complementarity provided by non-overlapping effect traits between species increases pollination," but the paper doesn't test this. Actually, you simply look for a correlation between functional divergence and yield. As far as I can tell, the studies included measure flower visitor communities and yield, but do not all directly measure pollination service. It has been shown that there can be a relationship between visitor communities and yield that is not mediated through pollination (Bartomeus et al (2014) PeerJ, 2: e328, which includes some of the same co-authors). Please be more accurate with the language, and clear about what you are actually measuring.

Lines 125-128: These sample size numbers seem bizarre to me. The sample size for a meta-analysis is usually the number of studies, not the number of recorded insects. Please report the number of published studies, or individual datasets used. From Tables 1 and 2, this is $n=7$ for mesocosm studies and $n=16$ for field studies. These are small sample sizes for meta-analysis.

Line 152: You introduce a term 'effect group richness' that is not defined thus far. Please define this in your introduction. It is not a standard term.

Line 181-183: the correlation between species richness and yield was only found across 7 mesocosm studies, so I feel this is overstated.

Line 241-243: "Complementarity contributed to increased yields under both conventional and hybrid breeding systems grown in different geographical regions and farming systems." You have not shown this. Only that functional divergence is correlated with increased yield.

Lines 243-245: Please suggest some management practices that would increase functional diversity of flower visitors. Is there any evidence that any of these practices actually enhance functional diversity?

A forest plot like Figure 1 would normally show the study identities. Please can you add?

Figure 2 needs to report the sample size (how many individual field studies, mesocosm studies)

Comments on methods:

Line 264: To simply state that "we identified studies" is not really acceptable in a published meta-analysis. By what method did you search for studies? What criteria did you use to select them? This is a very interesting analysis, but may suffer from having overlooked available datasets.

Line 280-282: "Where field experiments included exclusion cages these were treated as separate data points representing a zero pollinator abundance controls." Can you please provide some further information about this. How many of the studies does it apply to? It strikes me that it could be extremely important in generating the positive correlations, and I would like to know how important. Since the correlation between yield and abundance for a good number of the studies shown in figure 1 is not significantly different from 0, I want to know whether the studies with positive correlations are those that include a caged zero pollinators data point.

Line 329: should be representing?

Line 340: should be 'relevant'?

Line 393: "Mixed effects meta-analysis were used...". Do you mean mixed effects models? What were the random and fixed effects? Some basic methods information that should be in the main text is missing here.

Line 417-419: To ensure robustness, studies showing high levels of influence (Cook's distance > 1) on estimates of correlation coefficients were excluded (see Supplementary Methods 4). How many were excluded for this reason? This matters when the sample sizes are so small.

Line 587-588: "ME was instrumental in deriving behavioural trait metrics." What did he do exactly? This is vague.

Line 591: Should be Pearson's?

Reviewer #2:

Remarks to the Author:

General comments

This study uses a meta – analytical approach to estimate potential influences of structure and functional diversity of insects communities on crop yield, specifically in oilseed rape. Key results are important and relevant. Conclusions are original and would provide important information for implication of both oilseed and other crops management. Results presented are of immediate interest to many people working in several disciplines. Authors are dealing with crucial issues about ecosystem services and food provision. However, I have some general comments I consider authors would address before publication.

(1) Abstract starts mentioning 'the components of insect communities'. I consider is not a clear starting point for general readers. Even the manuscript does not uses too specific or technical concepts or terms, I suggest to make a final revision on how you define (and where in the text you define), all concepts used in the text. For instance, in line 78 authors mention 'the components of insect communities'. I suggest define earliest what they mean about 'components' (e.g. species richness, diversity, functional groups). They later mention community composition in line 84, and community structure in line 87. I guess both are part of these components, but this is not fully clear in the text. In line 94, authors mention that 'distinct pollinator communities', but in terms of structure, composition? The paragraph starts talking about community structure, but later mentions species-specific interactions, related also with composition.

(2) There is a bias to the Northern Hemisphere and specifically to Europe. Oilseed rape is also cultivated or produced in other regions of the world. Conclusions are original and would provide important information for implication of both oilseed and other crops management; results presented are of immediate interest to many people working in several disciplines, but they are much constrained only to a few regions in the Globe. Even I did not find flaws in the manuscript, authors should consider this geographical bias. I find conclusions and data interpretation are valid and reliable, but not fully robust for a global analysis of pollination services in oilseed rape.

(3) In the same reasoning, and consider both the low number of existing papers and the geographical bias, I would use a subtler language in the Abstract.

(4) About collinearity and pseudoreplication. In the main text they mention 181 experimental units for mesocosm (only 7 papers), and 222 in the field experiments (16 papers). Meta-analyses included yield metric as a random effect to account for methodological differences among studies, but what about using several data from the same study?

(5) Even the study uses data at community level (and not species) I'm not fully sure about the statistical implications of using several taxonomically related species. I know some phylogenetic approach to dealing with species in meta – analysis, but this is outside the scope of my expertise, and I'm unable to assess fully. I consider authors should mention how or why phylogeny is not a constraint in this study or if it is, how they deal with it.

Minor comments

L. 61. Authors explain what 'effects' traits means in lines 95-96, but in the Abstract this appears as a confusing term / concept, at least for those readers that are not used to pollination interactions. I would comment a while about what effect traits means.

L. 61. I would not use 'effect traits promote yields', even using Pearson correlations that resulted significant.

L. 70 I would use 'could be an effective strategy' instead 'is'

L. 85. Add 'the' before the word 'pollination'.

L. 95 Morphological and behavioural characteristics 'of the pollinator', right? How sounds saying like this: 'Morphological and behavioural characteristics of pollinators controlling these species-specific differences in their effectiveness are typically referred to as effect traits'.

L. 97. Add 'these' before the word 'effect'.

L. 98. I guess the last part of the sentence should be corrected (in singular). The sentence starts talking about the distribution (it remains), and not about the effects (they remain).

L. 106-108. You should give a reference for this.

L. 115-177 Please clarify... differences among effect traits..?

L. 117-118. I suggest moving this sentence to the final of the paragraph.

L. 121. 'whether pollination', measured as what? Or using which variable?

L. 265. Communities can be quantified, or characterized? Perhaps better being clear about which community specific variables you quantified.

L. 279 'a field' datum/plot? and, perhaps 'data on pollinators communities'

L. 326. Replace the second 'where' by were

L. 355. Add 'the' before the '15 effect traits'.

L. 624 Remove 'of' after 'on'

L. 626. Functional structure or divergence?

Tables 1 & 2 could be better placed in Supplemental Material

I suggest adding data and paper numbers in Figure 2 for each meta – analysis. In the main text they mention 181 experimental units for mesocosm (only 7 papers), and 222 in the field experiments (16 papers), but what are the results when sparing these numbers in each analysis (each moderator)?

In Table S2 caption, replace 'form' by 'from'

Reviewer #3:

Remarks to the Author:

The authors have set for themselves the ambitious task of evaluating the role of functional diversity in the pollination of an internationally cultivated crop, oilseed rape. They collected data from European, North American and Asian sources and modelled crop yield as a function of several different diversity metrics, including Functional Divergence (FDiv), functional group richness, species richness, abundance and Community Weighted Means (CWM). One of the strongest points to this manuscript is its readability. There is a unifying thread running through the presentation of objectives in the introduction, the analytical methods adopted to address those objectives, the results obtained, and the conclusions drawn from them. The authors were correct in asserting that few studies have assessed the role of the functional diversity of the insect community on crop pollination, and this appears to be a novel and important contribution to the literature. My main criticisms pertain to shortcomings in the results and their interpretation, suggesting to me that the narrative of this impressive work has yet to fully developed.

The authors have identified FDiv as a significant and positive predictor of crop yield across their sample of studies but were not able to explain why this would be so with the results obtained. When interpreting this finding, allusion was made on lines 205-211 to communities with high FDiv being active for a larger part of the flowering period due to differences in interspecific activity periods and responsiveness to environmental conditions, but this was simply an inference. No analytical comparison was made in pollinator abundance trends through time, or how their phenology compared with that of oilseed rape across studies. Instead, the main vehicle through which they might interpret the role of specific traits or, perhaps, trait combinations (CWM) is framed in terms of the mass-ratio hypothesis and found to yield insignificant results. Functional Divergence may then be important as an index of functional diversity and the supposed complementarity between different traits, but the mechanism by which complementarity arises in this study system is unresolved.

Perhaps the specifics of oilseed rape pollination were deemed inappropriate for the cross-continental scale of the study or the international readership of the journal. My concern, however, in not rooting the interpretation of FDiv in the science of insect-plant interactions is that the significant result obtained for this predictor variable may be a statistical artefact as opposed to biologically meaningful. From Figure 1, it appears as if this relationship (yield \sim FDiv) holds true for less than half of the studies examined (though, notably, this figure was not actually referenced in the manuscript) with considerable variance in its correlation. I also wonder about the reason for which FDiv was chosen as a metric, as many other indices of functional diversity exist and some (e.g. Functional Dispersion) with desirable properties over those of FDiv (see Laliberté and Legendre 2011). Was this the index that produced the best result? Without a clearer understanding of the biological interpretation of this index framed in terms of the distribution of traits that it models, and a justification of its use over similar indices, the reliability of this finding remains an open question.

My recommendation would be to further develop the analysis and better understand which complement of traits results in higher yield. The authors mention the possibility of using the Fourth Corner Analysis to do so, but that it was "impractical" given either the scope of the study or the data structure of the meta-analysis. Impractical is, of course, not the same thing as impossible and I would urge the authors to explore this avenue further should it provide insights into the interpretation of the main finding of this study. Alternative approaches using generalized linear or mixed models are proposed by Jamil et al. (2013) and Brown et al. (2014) (see below for full reference) that may be appropriate for the nested data structure of a meta analysis. Even a graphical projection of the ordinal trait space from which FDiv is derived showing the distribution of pollinators and their corresponding traits would help interpret this finding. Such assessments would only serve to strengthen the analysis and enrich the narrative being developed.

I have provided additional comments below that pertain to lines and sections of the manuscript. Of editorial importance, the authors have the tendency to use introductory clauses but not separate them from their corresponding dependent clauses with a comma. For example, on line 170, you write, "In all three cases the effect of CWM trait values explained no additional variance in yield above that predicted by the effect of abundance alone." Arguably, "In all three cases" is an introductory clause and should be followed by a comma. It may be a stylistic choice as the omission seems consistent throughout the manuscript, but I would recommend revising this decision as it does little to help with the readability of the text. I have noted some places where commas would be useful, but there may be more. On a related note, the entirety of Table 3 needs to be edited as the quality of the language and the formatting (e.g. erroneous use of superscript) used therein does not meet publication standards.

Line-by-line commentary

line 71 – Is this study really global in scope? The studies analyzed are all mostly in Europe.

line 77 - Insert comma before "of which"

line 78 - Change "the pollination service" to "pollination services"

line 82 - Comma after "Indeed" and after "climates"

line 85 - Change "service" plural

line 99-110 : Unclear what the pertinence of this paragraph is relative to the logic of the introduction. Does not appear to frame the subsequent research questions in a useful way, as it is focused on the difficulty of obtaining reliable effect trait data. Would be more useful to discuss the differences between complementarity and mass ratio hypotheses, or the mechanisms by which trait complementarity would promote pollination. It would also be essential to mention in the introduction the difference between mesocosm and field based studies, as the "Results" section opens with a mention of mesocosm findings without a prompt as to what this refers to. This is presuming that the reader does not move first to the "Methods" section, but the text should flow well enough as presented without having to jump sections. A mention of the utility of a meta-analysis approach may also be warranted.

Line 114 - "...context-specific effects of insect pollination on yield" is ambiguous in terms of its wording and does not help to clarify whether insect pollination is beneficial or not to crop yields. A clear statement on the utility of insect pollination for this crop is needed.

line 125 - Change "The data" to "Data" or rephrase the whole sentence as "The data set used in the meta-analysis was..."

line 125 - Change "meta-analyses" to "meta-analysis" as only a single meta-analysis was conducted.

line 126 - 56 taxonomic units listed but 57 mentioned (36+19+ two functional groups)- please clarify.

line 128 - "All taxonomic units were found..." Why is this essential information to report? I see that it mirrors what is said about the low species richness in the mesocosm studies, but it may be more interesting to detail the range in species richness values. At least that would flow more in line with the numerical summary offered in the previous and subsequent sentences.

line 139 - I really appreciate the different test statistics reported when describing the results of the models.

line 143- I wonder about the generalizability of the mesocosm finding. Do the results pertain only to species poor conditions? Were the same handful of species used across studies, so that "richness", in this case, reflects not really a property of the bee "community" but the effectiveness of particular species or even their interactions? It seems misleading to compare findings on richness between mesocosm and field settings when the total number of possible species is so different across the two.

line 144- Was rarefied richness used or raw (untransformed) richness values ? Calculating the projected richness values with the iNext package in R may be a useful means of controlling for the confounding effect of sampling bias across studies and sampling events.

line 153 - Insert comma after hypothesis.

line 156-158 : "...although this model included a significant positive effect abundance as a moderating

effect of this relationship" - awkward phrasing, please simplify. Something like "although this relationship was significantly moderated by...".

line 159 - Please standardize the way in which "conventional / hybrid" breeding type are mentioned in the text (compare with line 147). Consider pluralizing "breeding type" (i.e. hybrid or conventional breeding types).

line 159 - Insert "the" in the segment "as a moderator for the response of oilseed rape".

line 150 - 161 - I appreciate that the ordering of the terms "functional divergence" and "effect group richness" is consistent throughout this section.

line 164 - Insert comma after "success".

line 165 - I suspect that it should be "community weighted mean trait values" or "community weighted means" but not "community weighted means trait values".

line 170 - Inset comma after "cases".

line 175 - The term "moderator" is lacking a compliment ("moderator of something"). Either spell out what relationship was being moderated, as in line 159, or rephrase "acted as a moderator" to "moderating these relationships".

line 180 - Insert comma between "oilseed rape" and "the".

line 180-181 : I'm not convinced that this is the best way to start the discussion paragraph, especially if someone were to jump from the introduction to the discussion section. Is this the main finding of the article, and the one that would prompt the reader to continue through the discussion? The sentence should point to the "functional complementarity" of the pollinating communities described in the study. The species were described in terms of their numerous functional traits, so their lack of functional equivalency is unsurprising. As phrased, there is no indication that a lack of functional equivalency promotes or reduces yield, which should be made clear for those that skipped the results section.

line 181- 183: Despite beginning with "At the most basic level", this sentence is quite long and complicated, as it attempts to articulate three clauses linked by "that". Please use simpler and more direct language. Also, what do you mean by "idiosyncratic species effects"?

line 186- Insert comma after "studies".

line 189 - Insert comma after "model".

line 188 - 190 : This is an odd conclusion to draw about the limitation of mesocosm studies. Perhaps they do not fully "simulate" the competitive (e.g. low species richness and abundance) or environmental conditions of natural conditions, but I wouldn't say that they aren't "integrated". In fact, they may be better integrated because they are often controlled for, explicitly, in their study designs.

line 192 - The verb "to have" should conjugate with "number" and not "clusters".

line 203- I suppose some biological processes are continuous, but some traits are better modelled as discrete (e.g. presence or absence pollen bearing corbiculae). This in itself shouldn't be a reason for functional divergence modelling yield better than functional group richness, especially since species richness was found to be a strong predictor in the mesocosm studies. Functional group richness, arguably, follows the same logic as species richness in terms of its discreteness.

line 205 - The authors present functional divergence as the mechanism for how pollination is provided but do not provide any details about the actual mechanisms of that mechanism. Allusion is subsequently made to difference in activity periods that result in better overall coverage of the flowering period, however this was never directly modelled in the set of fifteen traits that they measured (lines 332-338). As a reader and reviewer of this article, I would be keenly interested to know more about why functional divergence is playing such a key role and in a consistent way across the study systems. More explanation is needed here.

line 228 - I'm confused why the fourth corner method was deemed 'impractical'. The practicality of a method isn't really grounds for not attempting it. One complication would be the nested structure of the data set (observations per study). An alternative approach would be to use the one proposed by Brown et al. 2014 or Jamil et al. 2013 (references below). It might be possible to model crop yield as

a function of pollinator abundances, their traits, and while using "study" or "yield metric" as a random effect to control for within study autocorrelation. Adopting such a method might help clarify why functional divergence was found to be predictive, beyond what was possible with CWM.

Brown, Alexandra M., et al. "The fourth-corner solution—using predictive models to understand how species traits interact with the environment." *Methods in Ecology and Evolution* 5.4 (2014): 344-352.

Jamil, T., Oz inga, W.A., Kley er, M. & ter Braak, C.J. (2013) Selecting traits that explain species–environment relationships: a generalized linear mixed model approach. *Journal of Vegetation Science*, 24, 988–1000.

line 245 - It's not immediately obvious how one would increase the functional diversity of the overall pollinator community. I can see how manipulating the functional diversity of plant communities might be possible, since you can choose which plants to plant, but pollinator communities would be much more difficult to manage at a farm scale. I presume the authors mean wild pollinators, as there aren't sufficient domesticated pollinators (honey bees, bumble bees, mason bees) to have substantial increases in FD. Existing management practices that are recommended (e.g. nest boxes, flower margins) would promote species richness, abundance and perhaps functional divergence, but would not target functional divergence in particular. I also wonder how it would be measured on a regular or practical basis to track the effectiveness of such approaches.

line 239 - This seems to be a reference to the findings presented in Figure 1, but I was not actually able to find any mention of these results in the results section of the text. In fact, Fig. 1 does not appear to be mentioned anywhere except for the Figures sections. Please revise accordingly.

line 260 - Insert comma after community.

line 266- Insert comma after resolution.

line 290 - Replace 7 with seven.

line 292 - Please outline where the mesocosm studies were conducted.

line 359 – Replace 5 with five.

line 364 - Please include information as to why functional divergence was used over other functional diversity metrics (e.g. functional dispersion). It's use should be justified, otherwise a comparison of its effectiveness in predicting yield may be warranted.

line 388 - Please include "a" after "In".

line 393 - Change "were" to "was".

line 393 (Statistical analysis section) – To what extent did you attempt to control for environmental factors while sampling across your studies? An example would be mean temperature, since presumably this was measured in many cases. Other environmental covariates of interest would be wind speed, cloudiness, humidity, time of day and time of year. These variables may covary with yield and bias the results, as would other variables that would affect the growing conditions of a crop in a given milieu (e.g. distance from field edge, mean temperatures throughout the growing period, mean precipitation rates, whether the field was organically or conventionally managed). If these variables seem pertinent to the authors, then they should be tested and accounted for.

line 590 - This figure is not presented in the main body of the manuscript!

line 591 - Delete "and" after abundance. Replace by comma.

line 598 - This is just a repetition of the title of the figure. Please revise for more original sentence structure or delete.

line 602- Insert comma after "insect abundance".

line 609 - It would be informative to include year of study in the table. It would also be helpful to have a map showing where all of the studies were conducted, to have an idea of its international scope. Although conclusions are drawn on the "global" ramifications of this meta-analysis (line 71), the studies are actually rather Euro-centric in their distribution and there's virtually no representation from the central hemisphere.

line 613 - Delete "include".

line 624- Change to "... are based on observations of wild pollinator communities under..."

Table 3

effect trait 1 - change "characterises" to "characterized"; change "affect" to "affects"

effect traits 2,3-4,5 and 6 need to be formatted so that the phrase "but augmented and additional with unpublished data" is not superscript. It also needs to be changed to simply "but augmented with unpublished data". Insert a space between the closed parenthesis and the word "based".

effect trait 7 - I had to reformat this whole paragraph. Please review and see where I made changes.

"Hairiness affects pollen grain deposition on stigmas 51 and, in bees, is used to detect electromagnetic fields emitted by flowers as pollination cues 66. For each species, body parts that contact oilseed stigmas (head, thorax, sternum, abdomen underside, femora, tibiae and meta-tarsus (legs assessed separately)) were scored as: 1) coarse setae or extremely short hairs; 2) short (c. basal tibiae 1 diameter) but dense hairs (>50 mm²); 3) long (>basal tibiae 1 diameter) dense (>50 mm²) hairs. This score was summed and given as a percentage of the maximum score of 24.

N.B. What is meant by "1 diameter"? Is a unit measure missing here?

effect trait 8- change "Tongue" to "The length of the tongue" or "The morphology of the tongue". Also change to "A separate category is listed for insects with chewing mouthparts".

effect traits 9 - 13 : please note the changes that I've made to the following paragraph:

The presence of setae specifically used to collect pollen, listed by Michener 52 as the basitarsal scopa (trait 9), femoral corbicula (trait 10), strict tibial corbicula (trait 11), propodeal corbicula (trait 12) or abdominal corbicula (trait 13). Note these structures are associated with bees, however, their absence will affect the pollen carrying capacity and thus likelihood of pollen stigmal transfer of other species (e.g. for flies).

effect trait 14: please note the changes that I've made to this section:

Pollen carried only in the crop and, as such, not available for pollination 52. As above, these structures are associated with bees, however, their absence may affect the likelihood of pollen contacting plant stigmas for other pollinating groups.

effect trait 15: Please note the changes in the following paragraph:

Pollen in corbicula storage structures may be either dry or moistened. Moistened pollen is less freely available for deposition onto plant stigmas 52. As above, these structures are associated with bees, however, their absence may affect the likelihood of pollen contacting plant stigmas for other pollinating groups.

Fig. 1 - The three panels need to be aligned on the lower horizontal axis. Each panel appears to have been narrowed individually so that the dots representing the mean values are more oval than circular. Ideally, the points would be circular. This whole figure was not mentioned in the text. An x-axis title would be helpful.

Fig. 2 - I really appreciate the formatting of this figure, as it's quite easy to interpret. However, It would be helpful to be presented with scatter plots or partial effect graphs for each predictor variable

to get a sense of the distribution of the data or the residuals. Perhaps this could be provided as Supplemental Material in addition to the funnel plots. It would be used to see whether the significant relationships are not being driven by a subset of studies. Moreover, the values beginning with $\mu =$ # [#, #] need to be aligned right.

Reviewer #1 (Remarks to the Author):

Comment: This paper presents an analysis of several studies of the statistical relationship between various measures of the flower visitor community structure and yield, for oilseed rape. Whilst it is a very useful and interesting approach, and provides original insight, there are some issues with how the paper is written. The theoretical background is not well explained, key details are missing from the methods and there are several careless typing errors.

Response: The paper has been considerably revised with substantial rewrites, clarifications and new analyses. We now include a more comprehensive paragraph in the introduction explaining the two key complementarity and mass ratio hypothesis that were tested. Following your comments and those of the other referees we have addressed the typos.

Comment: For it to be publishable in Nature Communications, it would need to be substantially revised to improve the quality of reporting and argument. Also, I have a concern about how the studies were identified. There may be other studies of oilseed rape pollination in the literature that are not included here, and the paper does not give me any confidence that the authors searched for them.

Response: The paper has been considerably revised with substantial rewrites, clarifications and new analyses. We include more detail in the main paper section as to the search criteria used in initial study selection. We now include in supplementary methods 3 a PRISMA flow diagram (<http://www.prisma-statement.org/>) outlining the full process of study selection and exclusion. PRISMA is an evidence-based minimum set of items for reporting in systematic reviews and meta-analyses.

Comment: I am uneasy that there is no critical discussion about whether relationships between flower visitor community structure and yield actually reflect pollination services. An unspecified number of the 16 field studies included exclusion cages (lines 280-282), and therefore were measuring a service provided by pollinators. But this is not true for all the studies. This really needs to be clarified.

Response: Following other comments below we now make which of the studies include exclusion cages (14 of the 16 field based studies) in table 2. We include additional text in the methodology justifying this approach. As an important caveat to the study we also make it clear that we are applying a correlative approach to identify relationships between metrics of community structure and yield. We include the following lines in the opening of the discussion to emphasise the limitations of the approach used.

Detailed comments:

Comment: Line 69 : Change to ‘management practices that increase pollinator abundance’

Response: changed as suggested

Comment: Line 96: introduction of the term ‘effect traits’ – I am not familiar with this term, so it would be good to have a reference here that defines it, or better still, a definition.

Response: We now include a definition of effect traits in the introduction at its first mention.

Comment: Lines 111-122: I don’t find this a very clear description of the research question, approach and hypotheses. As this is a meta-analysis, you should at least state here what type of studies you aimed to include, from what sources you searched for them. Then develop research questions and hypotheses to test with the resulting dataset. The text sort of does this, but it’s very vague. For instance: ‘using a meta-analysis approach’. What does this mean?

Response: We have added a more detailed section in response to your comments and those of ref#3 to explain the two key hypotheses being tested (mass ratio and complementarity). This should now give context to the results section (we originally described these in more detail in the methods, but the format of Nat. Comms means that this was not the place to do this as this comes after the discussion) We now go into considerably more detail in the methods section describing the process of study selection and include a PRISMA flow diagram (<http://www.prisma-statement.org/>) in Supplementary Methods 3 to describe in detail the approaches used including study inclusion and exclusion. In response to other comments from you we now clearly state in the opening lines of the results the number of studies used in the meta analyses (we think this is the better place), although make clear in the introduction the geographical bias of these studies. We also include a line in the final paragraph describing what a meta-analysis is used for.

Comment: There is very little explanation of the competing hypotheses being tested (complementary and mass ratio). Please expand on these, and explain them properly. For example, your results section implies that the complementarity hypothesis predicts a positive correlation between functional divergence and yield, but you haven’t really explained this in the introduction. There is a statement of this in the methods (line 347-349) but without references.

Response: Following this comment and those from ref#3 we now include in the introduction a new paragraph detailing both the complementarity and mass ratio hypotheses to give context to the rest of the paper.

Comment: The mass ratio hypothesis seems to come from plant ecology (reference 29-31). It seems oddly named and I miss an explanation of what it is, and why it might apply in this context.

Response: Yes, the origins of this theory are from plant ecology however it is relevant in the context of assessing any contribution of functional characteristics to the delivery of an ecosystem function. We now make this clear in the introduction. In response to your other comment (and an equivalent one from Rev#3) we now include a paragraph outlining the mass ratio and complementarity hypothesis in the introduction.

Comment: Lines 119-120, you claim to test “to what extent, (1) complementarity provided by non-overlapping effect traits between species increases pollination,”, but the paper doesn’t test this. Actually, you simply look for a correlation between functional divergence and yield. As far as I can tell, the studies included measure flower visitor communities and yield, but do not all directly measure pollination service. It has been shown that there can be a relationship between visitor communities and yield that is not mediated through pollination (Bartomeus et

al (2014) PeerJ, 2: e328, which includes some of the same co-authors). Please be more accurate with the language, and clear about what you are actually measuring.

Response: We now make it clear in the hypothesis section of the introduction and elsewhere throughout the manuscript that we infer pollination from correlation with increasing yield. WE state that we consider the effects of pollination only in terms of correlative relationships between pollinator community structure and oilseed rape yield. We also make clear in the introduction that pollination can have other benefits to crop quality that we do not consider, such as the seed oil content for oilseed rape.

Comment: Lines 125-128: These sample size numbers seem bizarre to me. The sample size for a meta-analysis is usually the number of studies, not the number of recorded insects. Please report the number of published studies, or individual datasets used. From Tables 1 and 2, this is $n=7$ for mesocosm studies and $n=16$ for field studies. These are small sample sizes for meta-analysis.

Response: We now clearly state that the meta-analysis sample sizes are $n=7$ for mesocosm studies and $n=16$ for field studies. WE make clear that these are the replicates used in the meta-analysis. However, we keep the other metrics as they are relevant in terms of providing the reader with an idea of the breath of the raw data from which these studies originate and now have relevance for additional analyses requested by reviewer 3.

Comment: Line 152: You introduce a term 'effect group richness' that is not defined thus far. Please define this in your introduction. It is not a standard term.

Response: We now define the term effect group here.

Comment: Line 181-183: the correlation between species richness and yield was only found across 7 mesocosm studies, so I feel this is overstated.

Response: Line now says 'However, this correlation between species richness and yield was only found for the limited number of mesocosm studies assessed in the meta-analysis.'

Comment: Line 241-243: "Complementarity contributed to increased yields under both conventional and hybrid breeding systems grown in different geographical regions and farming systems." You have not shown this. Only that functional divergence is correlated with increased yield.

Response: WE now make it clear though the manuscript that ultimately we are assessing only correlations. For example, in the new opening paragraph of the discussion we have the line 'As these relationships between pollinator community structure and yield were correlative this does not represent a direct experimental demonstration of the complementarity hypothesis. '

Comment: Lines 243-245: Please suggest some management practices that would increase functional diversity of flower visitors. Is there any evidence that any of these practices actually enhance functional diversity?

Response: We now include a range of example management practices in the conclusion.

Comment: A forest plot like Figure 1 would normally show the study identities. Please can you add?

Response: We now include number codes for each of the studies that link to the full descriptions of each study given in Tables 1 & 2.

Comment: Figure 2 needs to report the sample size (how many individual field studies, mesocosm studies)

Response: We now report sample sizes for this figure.

Comments on methods:

Comment: I have a concern about how the studies were identified. There may be other studies of oilseed rape pollination in the literature that are not included here, and the paper does not give me any confidence that the authors searched for them. &

Line 264: To simply state that “we identified studies” is not really acceptable in a published meta-analysis. By what method did you search for studies? What criteria did you use to select them? This is a very interesting analysis, but may suffer from having overlooked available datasets.

Response: We include more detail in the main paper section as to the search criteria used in initial study selection. We now include in supplementary methods 3 a PRISMA flow diagram (<http://www.prisma-statement.org/>) outlining the full process of study selection and exclusion. PRISMA is an evidence-based minimum set of items for reporting in systematic reviews and meta-analyses.

Comment: Line 280-282: “Where field experiments included exclusion cages these were treated as separate data points representing a zero pollinator abundance controls.” Can you please provide some further information about this. How many of the studies does it apply to? It strikes me that it could be extremely important in generating the positive correlations, and I would like to know how important. Since the correlation between yield and abundance for a good number of the studies shown in figure 1 is not significantly different from 0, I want to know whether the studies with positive correlations are those that include a caged zero pollinators data point.

Response: The use of zero abundance controls represent standard methodologies for assessing the contribution of pollinator communities to increasing crop yields and have been used for a wide range of crops and experimental design under both mesocosm and field conditions. Of the 16 field studies included in the analysis only two did not include zero abundance controls and so from the perspective of the meta-analysis these are an exception rather than the rule. We take your point and now clearly stated in the methods section which studies use exclusion cages to achieve a zero-abundance control with this being included as a new row in the study summary Table 2. Following on from another comment we now label studies in Fig. 1 (description of correlation coefficients for individual studies) so this can be linked to this information on which studies used zero-abundance controls.

Comment: Line 329: should be representing?

Response: changed as suggested

Comment: Line 340: should be ‘relevant’?

Response: changed as suggested

Comment: Line 393: “Mixed effects meta-analysis were used....”. Do you mean mixed effects models? What were the random and fixed effects? Some basic methods information that should be in the main text is missing here.

Response: The specific type of meta-analysis used, one that includes moderator effects such as the variety type classifier (hybrid vs conventional) is called a mixed effect meta-analysis (not my term, the one used in the literature). This is not to be confused with a general linear mixed model approach which as you say would have a specific random effect term. To clarify what this means we now include the line *‘In this type of meta-analysis correlations between pollinator community structure and yield are treated as random samples taken from a theoretical population, and from these summary correlation coefficients and measures of variance for that overall population are derived’*.

Comment: Line 417-419: To ensure robustness, studies showing high levels of influence (Cook’s distance > 1) on estimates of correlation coefficients were excluded (see Supplementary Methods 4). How many were excluded for this reason? This matters when the sample sizes are so small.

Response: The exclusion of outliers follows standard analytical procedure to avoid undue influence. While we see this as a key component of any robust meta-analysis we take the point of the reviewer it needs to be clear what was done. Although we felt that we had been upfront about this in the previous version detailing this in full in the Supl. Methods 4 (which the reviewer highlights) we now state explicitly for each analysis in the main paper how many outliers were excluded.

Comment: Line 587-588: “ME was instrumental in deriving behavioural trait metrics.” What did he do exactly? This is vague.

Response: WE have changed this to read ‘ME derived behavioural trait metrics.’

Comment: Line 591: Should be Pearson’s?

Response: The prevailing approach in the literature seems to be to use Pearson’s so I will use this.

Reviewer #2 (Remarks to the Author):

General reviewer summary: This study uses a meta – analytical approach to estimate potential influences of structure and functional diversity of insects communities on crop yield, specifically in oilseed rape. Key results are important and relevant. Conclusions are original and would provide important information for implication of both oilseed and other crops management. Results presented are of immediate interest to many people working in several disciplines. Authors are dealing with crucial issues about ecosystem services and food provision. However, I have some general comments I consider authors would address before publication.

Comment: (1) Abstract starts mentioning ‘the components of insect communities’. I consider is not a clear starting point for general readers. Even the manuscript does not uses too specific or technical concepts or terms, I suggest to make a final revision on how you define (and where in the text you define), all concepts used in the text. For instance, in line 78 authors mention ‘the components of insect communities’. I suggest define earliest what they mean

about 'components' (e.g. species richness, diversity, functional groups). They later mention community composition in line 84, and community structure in line 87. I guess both are part of these components, but this is not fully clear in the text. In line 94, authors mention that 'distinct pollinator communities', but in terms of structure, composition? The paragraph starts talking about community structure, but later mentions species-specific interactions, related also with composition.

Response: For the abstract we now open with the line '*How insects promote crop pollination remains poorly understood in terms of the relative contribution of overall abundance, species richness and the effect of functional differences between species*'. In the opening paragraph of the introduction after talking about 'components of community structure' we now clarify what we are referring to by the line '*Specifically the importance of overall community abundance, species richness, and the contribution of species specific functional behavioural and morphological characteristics that may act to facilitate pollination*'. In the rewrite the distinct pollinator communities line is now removed.

Comment: (2) There is a bias to the Northern Hemisphere and specifically to Europe. Oilseed rape is also cultivated or produced in other regions of the world. Conclusions are original and would provide important information for implication of both oilseed and other crops management; results presented are of immediate interest to many people working in several disciplines, but they are much constrained only to a few regions in the Globe. Even I did not find flaws in the manuscript, authors should consider this geographical bias. I find conclusions and data interpretation are valid and reliable, but not fully robust for a global analysis of pollination services in oilseed rape.

Response: In the abstract we have now removed the word 'global' from the final line reflecting your comments. The low word count of the abstract makes it hard to include more detail here, but tables 1 and 2 clearly define the geographical scope. However, we now make the geographical bias clear throughout the manuscript.

Comment: (3) In the same reasoning, and consider both the low number of existing papers and the geographical bias, I would use a subtler language in the Abstract.

Response: Deleted the word globally from the end of the abstract (although feel its still reasonable to say it's a globally important crop). In the conclusion we now include the line 'These results are, however, based on a data set biased to the Northern Hemisphere, and in particular Europe, and as such may not necessarily be generalisable to other regions where the crop is grown or interacts with functionally different pollinator communities.'

Comment: (4) About collinearity and pseudoreplication. In the main text they mention 181 experimental units for mesocosm (only 7 papers), and 222 in the field experiments (16 papers). Meta-analyses included yield metric as a random effect to account for methodological differences among studies, but what about using several data from the same study?

Response: Yes data from the same study are used but in each case their constituent data points represent independent farms or fields. We treat them as independent to account for the variety differences sown on sites which represents a potential factor likely to affect the description of the response between yield and metrics of pollinator community structure (note this is also accounted for in the models by the inclusion of the hybrid/conventional moderator). Inclusion of a study moderator as suggested becomes redundant when the moderator of yield metric is included in the analysis (which we consider to be a far more significant factor that needs to be accounted for) as it to a lesser or larger extent classifies the same thing (sites from the same study tend to use the same metric). Note that is we drop

yield metric as a moderator and include a study moderator it does not qualitatively change the results.

Comment: (5) Even the study uses data at community level (and not species) I'm not fully sure about the statistical implications of using several taxonomically related species. I know some phylogenetic approach to dealing with species in meta – analysis, but this is outside the scope of my expertise, and I'm unable to assess fully. I consider authors should mention how or why phylogeny is not a constraint in this study or if it is, how they deal with it.

Response: There is an inherent underlying link between trait diversity and phylogeny, and in this has been used as a measure of functional diversity (see use of phylogenetic measures as surrogates for diversity by (Petchey & Gaston 2006). This reflects the fact that phylogenetically distinct species tend to also be functionally distinct. Such links tend to focus more on the breath of all functional characteristics of a species (response and effect traits), and so are not necessarily relevant to the effects trait approach considered here (i.e. specific traits that affect a species capacity to influence pollination). Following your comment we now include a measure of Phylogenetic diversity (mean pairwise phylogenetic distance) which we test directly within the meta-analysis to see whether phylogenetic diversity characteristics of communities alone explain diversity trends.

Minor comments

Comment: L. 61. Authors explain what 'effects' traits means in lines 95-96, but in the Abstract this appears as a confusing term / concept, at least for those readers that are not used to pollination interactions. I would comment a while about what effect traits means.

Response: Replaced by the line 'We used a meta-analysis to test for correlations between these factors and the yield of a globally important crop, oilseed rape.'

Comment: L. 61. I would not use 'effect traits promote yields', even using Pearson correlations that resulted significant.

Response: Replaced by the line 'We used a meta-analysis to test for correlations between these factors and the yield of a globally important crop, oilseed rape.'

Comment: L. 70 I would use 'could be an effective strategy' instead 'is'

Response: changed as suggested.

Comment: L. 85. Add 'the' before the word 'pollination'.

Response: Following comment from ref#3 we pluralised services in this sentences so that 'the' is not required.

Comment: L. 95 Morphological and behavioural characteristics 'of the pollinator', right? How sounds saying like this: 'Morphological and behavioural characteristics of pollinators controlling these species-specific differences in their effectiveness are typically referred to as effect traits'.

Response: Changed as suggested.

Comment: L. 97. Add 'these' before the word 'effect'.

Response: changed

Comment: L. 98. I guess the last part of the sentence should be corrected (in singular). The sentence starts talking about the distribution (it remains), and not about the effects (they

remain).

Response: Sentence changed to read ‘The distribution of these effect traits within a pollinator community is expected to have a pivotal role in overall pollination services 3,7,20,21, however, it has often proved hard to empirically demonstrate this relationship.’

Comment: L. 106-108. You should give a reference for this.

Response: Added.

Comment: L. 115-177 Please clarify... differences among effect traits..?

Response: Sentence restructured for clarity.

Comment: L. 117-118. I suggest moving this sentence to the final of the paragraph.

Response: Moved.

Comment: L. 121. ‘whether pollination’, measured as what? Or using which variable?

Response: We clarify this in the line before as ‘...as inferred from positive correlations with increased yield of oilseed rape’

Comment: L. 265. Communities can be quantified, or characterized? Perhaps better being clear about which community specific variables you quantified.

Response: We have now expanded on this section to describe what we mean in terms of quantified. We highlight as well that this is described in greater detail in a following section (section ‘Pollinator Communities’). Note that the process of study selection is now clarified by the inclusion of a PRISMA flow diagram in supplementary methods 3.

Comment: L. 279 ‘a field’ datum/plot? and, perhaps ‘data on pollinators communities’

Response: Changed as suggested

Comment: L. 326. Replace the second ‘where’ by were

Response: Changed as suggested

Comment: L. 355. Add ‘the’ before the ‘15 effect traits’.

Response: Changed as suggested

Comment: L. 624 Remove ‘of’ after ‘on’

Response: Changed as suggested

Comment: L. 626. Functional structure or divergence?

Response: Changed to divergence.

Comment: Tables 1 & 2 could be better placed in Supplemental Material

Response: I take your point on this one, but other reviewers are very keen to add more references to individual studies (e.g in Fig.1) and I feel that without these tables their would be little context to these additions. I am going to leave them in as a main table for the moment for this reason, although if you still feel strongly about this I will remove them to the suppl. Material.

Comment: I suggest adding data and paper numbers in Figure 2 for each meta – analysis. In the main text they mention 181 experimental units for mesocosm (only 7 papers), and 222 in the field experiments (16 papers), but what are the results when sparing these numbers in

each analysis (each moderator)?

Response: We assume you are referring to Fig 1 as Fig2 are summary correlation derived from the meta-analysis and do not relate to individual studies directly. For fig 1 we now state which study the data originates from. Note we also now include raw correlation for all tested response at a site level (following request by rev#3) and include these in supplementary fig.1.

Comment: In Table S2 caption, replace 'form' by 'from'

Response: Changed

Reviewer #3 (Remarks to the Author):

Overall reviewer summary: The authors have set for themselves the ambitious task of evaluating the role of functional diversity in the pollination of an internationally cultivated crop, oilseed rape. They collected data from European, North American and Asian sources and modelled crop yield as a function of several different diversity metrics, including Functional Divergence (FDiv), functional group richness, species richness, abundance and Community Weighted Means (CWM). One of the strongest points to this manuscript is its readability. There is a unifying thread running through the presentation of objectives in the introduction, the analytical methods adopted to address those objectives, the results obtained, and the conclusions drawn from them. The authors were correct in asserting that few studies have assessed the role of the functional diversity of the insect community on crop pollination, and this appears to be a novel and important contribution to the literature. My main criticisms pertain to

shortcomings in the results and their interpretation, suggesting to me that the narrative of this impressive work has yet to fully developed.

Response: see specific comments below but we have implemented the majority of your comments, particularly those relating to the need for further analysis to pull apart the contribution of different traits to the effect of functional divergence on yield. This has allowed us to develop the discussion to consider in more detail the mechanistic contribution of functional traits to yield gains in this crop.

General: The authors have identified FDiv as a significant and positive predictor of crop yield across their sample of studies but were not able to explain why this would be so with the results obtained. When interpreting this finding, allusion was made on lines 205-211 to communities with high FDiv being active for a larger part of the flowering period due to differences in interspecific activity periods and responsiveness to environmental conditions, but this was simply an inference. No analytical comparison was made in pollinator abundance trends through time, or how their phenology compared with that of oilseed rape across studies. Instead, the main vehicle through which they might interpret the role of specific traits or, perhaps, trait combinations (CWM) is framed in terms of the mass-ratio hypothesis and found to yield insignificant results. Functional Divergence may then be important as an index of functional diversity and the supposed complementarity between different traits, but the mechanism by which complementarity arises in this study system is unresolved.

Perhaps the specifics of oilseed rape pollination were deemed inappropriate for the cross-continental scale of the study or the international readership of the journal. My concern, however, in not rooting the interpretation of FDiv in the science of insect-plant interactions is that the significant result obtained for this predictor variable may be a statistical artefact as opposed to biologically meaningful. From Figure 1, it appears as if this relationship (yield ~

FDiv) holds true for less than half of the studies examined (though, notably, this figure was not actually referenced in the manuscript) with considerable variance in its correlation. My recommendation would be to further develop the analysis and better understand which complement of traits results in higher yield. The authors mention the possibility of using the Fourth Corner Analysis to do so, but that it was “impractical” given either the scope of the study or the data structure of the meta-analysis. Impractical is, of course, not the same thing as impossible and I would urge the authors to explore this avenue further should it provide insights into the interpretation of the main finding of this study. Alternative approaches using generalized linear or mixed models are proposed by Jamil et al. (2013) and Brown et al. (2014) (see below for full reference) that may be appropriate for the nested data structure of a meta analysis. Even a graphical projection of the ordinal trait space from which FDiv is derived showing the distribution of pollinators and their corresponding traits would help interpret this finding. Such assessments would only serve to strengthen the analysis and enrich the narrative being developed.

Response: We take your point that there is potentially a need additional clarification of what traits contribute to the importance of Functional divergence in predicting yield and identified in the meta-analysis (see comment in following two paragraphs about why 4th corner and the Jamil GLMM equivalent are not appropriate). Taking a recommendation for the Jamil *et al.* (2013) we apply approaches suggested by Sonnier, Shipley and Navas (2010) where we focus on the specific question of identifying associations between traits and clusters in this case yield. This involves combining a sites x species table and species x traits table into a sites x traits table (in practice this produces a table of sites x CWM trait values). Then using glmm and appropriate random effects we can then look to see how this derived table can predict yield, producing a linear combination of effect traits (CWM) that predict yield. We present plots of the relationships for individual studies where the CWM predict yield with a variable importance score >0.25 in the new Supplementary Fig. 1 based on the information theoretic approach used for model simplification.

N.b. see response to your comment below about the use of 4th corner and equivalent GLMM based approaches.

General: I also wonder about the reason for which FDiv was chosen as a metric, as many other indices of functional diversity exist and some (e.g. Functional Dispersion) with desirable properties over those of FDiv (see Laliberté and Legendre 2011). Was this the index that produced the best result? Without a clearer understanding of the biological interpretation of this index framed in terms of the distribution of traits that it models, and a justification of its use over similar indices, the reliability of this finding remains an open question.

Response: the choice of ‘functional diversity’ index was not random, but was chosen to specifically reflect the question of complementarity in effect traits. In contrast to many functional diversity indexes functional divergence (FDiv) provides a much more obvious measure of complementarity as it measures spread across trait space and so directly relates to the concept of complementarity. Functional divergence is low when most individuals in a community have traits that are near the centre of functional trait space, and it greatest when individuals are mostly positioned at the edges of the functional trait space, a situation in which functional complementarity would be high. This is a unique property of this specific index and this was the reason for its choice. Note in the methods section we now specifically justify the use of the FDiv index with ‘*The complementarity hypothesis assumes that communities with non-overlapping trait distributions will be more likely to promote*

increased pollination (Hoehn et al. 2008; Garibaldi et al. 2011; Gagic et al. 2015). Functional divergence describes the extent to which species are either clumped or spread out in trait space (Mason et al. 2005) and as such represents a relevant metric for assessing complementarity. Other common diversity indexes, such as Rao's or FDiv, do not have this property. Functional divergence is low when most individuals in a community have traits near the centre of functional trait space and is greatest when individuals are positioned at the edges of the trait space',

Comment: My recommendation would be to further develop the analysis and better understand which complement of traits results in higher yield. The authors mention the possibility of using the Fourth Corner Analysis to do so, but that it was “impractical” given either the scope of the study or the data structure of the meta-analysis. Impractical is, of course, not the same thing as impossible and I would urge the authors to explore this avenue further should it provide insights into the interpretation of the main finding of this study. Alternative approaches using generalized linear or mixed models are proposed by Jamil et al. (2013) and Brown et al. (2014) (see below for full reference) that may be appropriate for the nested data structure of a meta analysis. Even a graphical projection of the ordinal trait space from which FDiv is derived showing the distribution of pollinators and their corresponding traits would help interpret this finding. Such assessments would only serve to strengthen the analysis and enrich the narrative being developed.

Response: See preceding response about the new analysis implemented to pull apart the mechanistic contributions of different traits to yield as mediated through the functional divergence effect identified in the meta-analysis. Note, I have further looked into the possibility of using 4th corner analysis. It is not in fact suitable for the analysis to be undertaken here. It is used to assess how the environment filters certain species traits. For example, whether the environment selects for species with rapid growth rates. This is useful when looking at the effect of environment on response traits that define sensitivity to environmental conditions. However, this is not the case here where we are looking at how effects traits (i.e. those likely to have an effect on pollination) impact on an emergent property of the crop, specifically yield. Conceptually it is not designed to test such an association, i.e. how traits, expressed by species affect yield (which would effectively be an environmental condition). Note, that by definition we could not flip around the env and trait matrices in the analysis as they describe different factors (e.g. site x env, and species x trait respectively). In the text we have now removed reference to fourth corner analysis. Similarly the approach you suggest by Jamil *et al.* (2013) from the ter Braak group is attempting to do a very similar thing (indeed they specifically state that they are a predictive alternative to 4th corner), just using a different computational approach based on mixed models as opposed to ordination based methods. Again this is used ‘... to identify which species traits and environmental variables best explain the species distribution, and which traits are significantly correlated with environmental variables.’. As such from the perspective of the current analysis this has the same issues as 4th corner analysis in that it is focusing on how the environment modulates specific assemblages through traits. Contextually we would be looking at how species through traits modulate env (in the context of this environment would be OSR yield).

N.b. see response above about the GLMM analysis now implemented to effectively address this point, i.e. picking apart the additive contribution of multiple effects traits in terms of the functional divergence response.

Comment: I have provided additional comments below that pertain to lines and sections of the manuscript. Of editorial importance, the authors have the tendency to use introductory clauses but not separate them from their corresponding dependent clauses with a comma. For example, on line 170, you write, "In all three cases the effect of CWM trait values explained no additional variance in yield above that predicted by the effect of abundance alone." Arguably, "In all three cases" is an introductory clause and should be followed by a comma. It may be a stylistic choice as the omission seems consistent throughout the manuscript, but I would recommend revising this decision as it does little to help with the readability of the text. I have noted some places where commas would be useful, but there may be more.

Response: we have now gone through and added apostrophise where appropriate to indicate introductory clauses.

Comment: On a related note, the entirety of Table 3 needs to be edited as the quality of the language and the formatting (e.g. erroneous use of superscript) used therein does not meet publication standards.

Response: In response to this comments and the comments below this table has now been changed.

Line-by-line commentary

Comment: line 71 – Is this study really global in scope? The studies analyzed are all mostly in Europe.

Response: We maintain that the crop is of global importance and that this is appropriate for the abstract. However, remove the 'global' from the final line. In the Methods we now make it clear that the range of studies is from the Northern Hemisphere and principally Europe.

Comment: line 77 - Insert comma before "of which"

Response:

Comment: line 78 - Change "the pollination service" to "pollination services"

Response:

Comment: line 82 - Comma after "Indeed" and after "climates"

Response:

Comment: line 85 - Change "service" plural

Response:

Comment: line 99-110 : Unclear what the pertinence of this paragraph is relative to the logic of the introduction. Does not appear to frame the subsequent research questions in a useful way, as it is focused on the difficulty of obtaining reliable effect trait data. Would be more useful to discuss the differences between complementarity and mass ratio hypotheses, or the mechanisms by which trait complementarity would promote pollination. It would also be essential to mention in the introduction the difference between mesocosm and field based studies, as the "Results" section opens with a mention of mesocosm findings without a prompt as to what this refers to. This is presuming that the reader does not move first to the "Methods" section, but the text should flow well enough as presented without having to jump sections. A mention of the utility of a meta-analysis approach may also be warranted.

Response: We now delete this paragraph and replace it with a new paragraph discussing aspects of the mass ratio and complementarity hypotheses, parts of which are now removed from the original methods section (where more detail on these hypothesis was previously provided). We mention the difference between mesocosm and field based studies to set the context for the results. We also state why meta-analysis are useful.

Comment: Line 114 - "...context-specific effects of insect pollination on yield" is ambiguous in terms of its wording and does not help to clarify whether insect pollination is beneficial or not to crop yields. A clear statement on the utility of insect pollination for this crop is needed.

Response: This line is replaced to read 'Although partially wind pollinated, studies have identified positive effects of insect pollination on yield'. We feel that this changed wording does set the context for the ambiguous partially wind pollinated nature of this crop.

Comment: line 125 - Change "The data" to "Data" or rephrase the whole sentence as "The data set used in the meta-analysis was..."

Response: changed.

Comment: line 125 - Change "meta-analyses" to "meta-analysis" as only a single meta-analysis was conducted.

Response: changed.

Comment: line 126 - 56 taxonomic units listed but 57 mentioned (36+19+ two functional groups)- please clarify.

Response: typo corrected to read 57

Comment: line 128 - "All taxonomic units were found..." Why is this essential information to report? I see that it mirrors what is said about the low species richness in the mesocosm studies, but it may be more interesting to detail the range in species richness values. At least that would flow more in line with the numerical summary offered in the previous and subsequent sentences.

Response: We delete this line. Note we state that across all studies only seven species were used to create the artificial mesocosm communities, with no individual study combining more than three species. This gives context to the smaller species pool used in the mesocosm studies.

Comment: line 139 - I really appreciate the different test statistics reported when describing the results of the models.

Response: No response required (we will keep these test statistics).

Comment: line 143- I wonder about the generalizability of the mesocosm finding. Do the results pertain only to species poor conditions? Were the same handful of species used across studies, so that "richness", in this case, reflects not really a property of the bee "community" but the effectiveness of particular species or even their interactions? It seems misleading to compare findings on richness between mesocosm and field settings when the total number of possible species is so different across the two.

Response: This is a fair point and the reason why we analyse these two studies separately. Mesocosms are useful in their capacity to manipulate communities completely, with exact information on abundance and species composition. This is not the case in field studies where ultimately we have only a sample of what is there, but not an exact definition. The obvious flip side of this is that mesocosms are limited due to their artificial and highly simplistic

nature in terms of the breadth of species composition. Treated separately the inclusion of both studies is relevant. We already discuss the points you raised in the opening paragraph of the discussion, but now include the lines *'Typically, mesocosms studies were composed of similar species (i.e. those suitable for captive rearing) and as such the response of yield may represent a special case resulting from a sub-set of species interactions not necessarily generalisable to those of more complex communities found under field conditions.'* Note, we already make it clear at the start of the results that the mesocosms are limited in species range, see line *'Across all studies only seven species were used to create the artificial mesocosm communities, with no individual study combining more than three species'*.

Comment: line 144- Was rarefied richness used or raw (untransformed) richness values ? Calculating the projected richness values with the Next package in R may be a useful means of controlling for the confounding effect of sampling bias across studies and sampling events.

Response: We do not use rarefied species richness, nor do we think it is appropriate for this kind of study. Rarefaction was an approach developed to standardise huge inventories of taxa collecting 1000's of individual. With sufficient numbers of individuals the rarefaction curve starts to level off and makes for reasonable comparisons between sites. However, where the number of individual counted is small rarefaction to the lowest common number of individual per site means that the discriminatory precision of this method is very poor (i.e. you are rarefying a the steepest point of the curve). This tends to give very unreliable artefact corrections of SR. The alternative approach you suggest of extrapolation has the same problems with this kind of data, its simply not rich enough in terms of the number of individuals to give a meaningful prediction of the extrapolated curve. Given that each site used a standard methodology and is treated as independent within the meta-analysis (or classified using a random effect in the new glmm, see below) we feel that the current approach is appropriate for the data.

Comment: line 153 - Insert comma after hypothesis.

Response: changed.

Comment: line 156-158 : "...although this model included a significant positive effect abundance as a moderating effect of this relationship" - awkward phrasing, please simplify. Something like "although this relationship was significantly moderated by..."

Response: changed as suggested.

Comment: line 159 - Please standardize the way in which "conventional / hybrid" breeding type are mentioned in the text (compare with line 147). Consider pluralizing "breeding type" (i.e. hybrid or conventional breeding types).

Response: We now standardise in the results to 'hybrid or conventional breeding types'

Comment: line 159 - Insert "the" in the segment "as a moderator for the response of oilseed rape".

Response: Changed.

Comment: line 150 - 161 - I appreciate that the ordering of the terms "functional divergence" and "effect group richness" is consistent throughout this section.

Response: no change required

Comment: line 164 - Insert comma after "success".

Response: Changed

Comment: line 165 - I suspect that it should be "community weighted mean trait values" or "community weighted means" but not "community weighted means trait values".

Response: Changed

Comment: line 170 – Inset comma after "cases".

Response: Changed

Comment: line 175 - The term "moderator" is lacking a compliment ("moderator of something"). Either spell out what relationship was being moderated, as in line 159, or rephrase "acted as a moderator" to "moderating these relationships".

Response: This has now been changed to include 'for these relationships.'

Comment: line 180 - Insert comma between "oilseed rape" and "the".

Response: Changed

Comment: line 180-181 : I'm not convinced that this is the best way to start the discussion paragraph, especially if someone were to jump from the introduction to the discussion section. Is this the main finding of the article, and the one that would prompt the reader to continue through the discussion? The sentence should point to the "functional complementarity" of the pollinating communities described in the study. The species were described in terms of their numerous functional traits, so their lack of functional equivalency is unsurprising. As phrased, there is no indication that a lack of functional equivalency promotes or reduces yield, which should be made clear for those that skipped the results section.

Response: We now open the discussion with a new introductory paragraph that summarises the key results.

Comment: line 181- 183: Despite beginning with "At the most basic level", this sentence is quite long and complicated, as it attempts to articulate three clauses linked by "that". Please use simpler and more direct language. Also, what do you mean by "idiosyncratic species effects"?

Response: following revision sentence is no longer in paper.

Comment: line 186- Insert comma after "studies".

Response: Changed

Comment: line 189 - Insert comma after "model".

Response: Changed

Comment: line 188 - 190 : This is an odd conclusion to draw about the limitation of mesocosm studies. Perhaps they do not fully "simulate" the competitive (e.g. low species richness and abundance) or environmental conditions of natural conditions, but I wouldn't say that they aren't "integrated". In fact, they may be better integrated because they are often controlled for, explicitly, in their study designs.

Response: Changed to read 'However, as mesocosms experiments typically are designed to control for confounding factors, including abundance, they do provide useful mechanistic insights into the importance of factors like species richness. Such effects may be harder to detect under field conditions, where community structure is not exactly known but rather estimated by what are ultimately limited sampling regimes.'

Comment: line 192 - The verb "to have" should conjugate with "number" and not "clusters."

Response: changed to 'has'

Comment: line 203- I suppose some biological processes are continuous, but some traits are better modelled as discrete (e.g. presence or absence pollen bearing corbiculae). This in itself shouldn't be a reason for functional divergence modelling yield better than functional group richness, especially since species richness was found to be a strong predictor in the mesocosm studies. Functional group richness, arguably, follows the same logic as species richness in terms of its discreteness.

Response: Yes its true that some traits are modelled better by categorical factors, but this is not what effect group richness is doing. By definition it clusters species into groups with higher within than out of group similarity based on a range of categorical and continuous traits. These individual clusters still have within group trait variation that is effectively ignored. Functional dispersion in contrast does not assume a threshold at which differences in traits are ignored and so has better resolution for identifying subtly trends linked to complementarity. Functional richness and species richness are as you say similar in terms of being discrete, but the difference is that species richness assumes all species are equally different, effect group richness effectively considers 'functional species' by sinking real taxonomic units into sub groups that do approximately (but not totally) the same thing). We now talk about this distinction to a greater level in the discussion.

Comment: line 205 - The authors present functional divergence as the mechanism for how pollination is provided but do not provide any details about the actual mechanisms of that mechanism. Allusion is subsequently made to difference in activity periods that result in better overall coverage of the flowering period, however this was never directly modelled in the set of fifteen traits that they measured (lines 332-338). As a reader and reviewer of this article, I would be keenly interested to know more about why functional divergence is playing such a key role and in a consistent way across the study systems. More explanation is needed here.

Response: Firstly we have implemented your suggestions about the need for an additional analysis (see comments above) to identify what traits are explaining yield using a glmm based approach. This is now a fundamental addition to the meta-analysis and addresses your main point. These results are now discussed in the discussion. We now remove from the discussion the example we previously had used linked to overlapping activity periods.

Comment: line 228 - I'm confused why the fourth corner method was deemed 'impractical'. The practicality of a method isn't really grounds for not attempting it. One complication would be the nested structure of the data set (observations per study). An alternative approach would be to use the one proposed by Brown et al. 2014 or Jamil et al. 2013 (references below). It might be possible to model crop yield as a function of pollinator abundances, their traits, and while using "study" or "yield metric" as a random effect to control for within study autocorrelation. Adopting such a method might help clarify why functional divergence was found to be predictive, beyond what was possible with CWM.

Response: Note, I having further looked into the possibility of using 4th corner analysis. It is not in fact suitable for the analysis to be undertaken here. It is used to assess how the environment filters certain species traits. For example, whether the environment selects for species with rapid growth rates. This is useful when looking at the effect of environment on response traits that define sensitivity to environmental conditions. However, this is not the case here where we are looking at how effects traits (i.e. those likely to have an effect on

pollination) impact on an emergent property of the crop, specifically yield. Conceptually it is not designed to test such an association, i.e. how traits, expressed by species affect yield (which would effectively be an environmental condition). Note, that by definition we could not flip around the env and trait matrices in the analysis as they describe different factors (e.g. site x env, and species x trait respectively). In the text we have now removed reference to forth corner analysis. Similarly the approach you suggest by Jamil *et al.* (2013) from the ter Braak group is attempting to do a very similar thing (indeed they specifically state that they are a predictive alternative to 4th corner), just using a different computational approach based on mixed models as opposed to ordination based methods. Again this is used '*... to identify which species traits and environmental variables best explain the species distribution, and which traits are significantly correlated with environmental variables.*'. As such from the perspective of the current analysis this has the same issues as 4th corner analysis in that it is focusing on how the environment modulates specific assemblages through traits. Contextually we would be looking at how species through traits modulate env (in the context of this environment would be OSR yield).

N.b. see response above about the GLMM analysis now implemented to effectively address this point, i.e. picking apart the additive contribution of multiple effects traits in terms of the functional divergence response.

Comment: line 245 - It's not immediately obvious how one would increase the functional diversity of the overall pollinator community. I can see how manipulating the functional diversity of plant communities might be possible, since you can choose which plants to plant, but pollinator communities would be much more difficult to manage at a farm scale. I presume the authors mean wild pollinators, as there aren't sufficient domesticated pollinators (honey bees, bumble bees, mason bees) to have substantial increases in FD. Existing management practices that are recommended (e.g. nest boxes, flower margins) would promote species richness, abundance and perhaps functional divergence, but would not target functional divergence in particular. I also wonder how it would be measured on a regular or practical basis to track the effectiveness of such approaches.

Response: We now define a selection of practices that could be used to achieve this. Note we link this in to your new suggested analysis used to pick apart the individual traits that feed into functional divergence.

Comment: line 239 - This seems to be a reference to the findings presented in Figure 1, but I was not actually able to find any mention of these results in the results section of the text. In fact, Fig. 1 does not appear to be mentioned anywhere except for the Figures sections. Please revise accordingly.

Response: Fig 1 shows the range of individual study effect sizes for the three key community metrics of abundance, species richness and functional divergence correlation with yield. We now make sure these are referenced in the text.

Comment: line 260 - Insert comma after community.

Response: changed.

Comment: line 266- Insert comma after resolution.

Response: changed.

Comment: line 290 - Replace 7 with seven.

Response: changed.

Comment: line 292 - Please outline where the mesocosm studies were conducted.

Response: added.

Comment: line 359 – Replace 5 with five.

Response: changed.

Comment: line 364 - Please include information as to why functional divergence was used over other functional diversity metrics (e.g. functional dispersion). Its use should be justified, otherwise a comparison of its effectiveness in predicting yield may be warranted.

Response: See comments above. The Methods section now states specifically why we use the functional divergence index justifying its use for the assessment of complementarity. *‘The complementarity hypothesis assumes that communities with non-overlapping trait distributions will be more likely to promote increased pollination (Hoehn et al. 2008; Garibaldi et al. 2011; Gagic et al. 2015). Functional divergence describes the extent to which species are either clumped or spread out in trait space (Mason et al. 2005) and as such represents a relevant metric for assessing complementarity. Other common diversity indexes, such as Rao’s or FDiv, do not have this property. Functional divergence is low when most individuals in a community have traits near the centre of functional trait space and is greatest when individuals are positioned at the edges of the trait space’.*

Comment: line 388 - Please include "a" after "In".

Response: Changed.

Comment: line 393 - Change “were” to “was”.

Response: Changed.

Comment: line 393 (Statistical analysis section) – To what extent did you attempt to control for environmental factors while sampling across your studies? An example would be mean temperature, since presumably this was measured in many cases. Other environmental covariates of interest would be wind speed, cloudiness, humidity, time of day and time of year. These variables may covary with yield and bias the results, as would other variables that would affect the growing conditions of a crop in a given milieu (e.g. distance from field edge, mean temperatures throughout the growing period, mean precipitation rates, whether the field was organically or conventionally managed). If these variables seem pertinent to the authors, then they should be tested and accounted for.

Response: We lack continuous measures of the kind of variables you are talking about across the studies. In terms of when the pollinators are monitored some studies do record information on weather covariates, other simply work on a threshold basis of conditions (typically those specified by Pollard & Yates) and record when those conditions are met. Even if this were not the case weather conditions recorded for pollinators by defenitoin reflect the ‘good’ weather days when people go out and record pollinators, they do not reflect local or regional weather trends. Such trends were not recorded. The biggest factors likely to result in between study variation are linked to oilseed variety type (we account for this with our hybrid /conventional moderator) and what the yield metric was (again accounted for by a moderator. Further the input for the meta-analysis is based on Pearsons correlation coef and so does not have the capacity for inclusions of other covariates. Even if this was possible the absence of consistent between study covariates would mean that if tried different components of unexplained variation would be removed from each study depending on what covariates were available) creating a confounding effect. Ultimately we are looking for consistent

between study correlations in the meta-analysis, within which we have accounted for the likely biggest factors affecting yield response as moderators. Note there is no such thing as organic oilseed rape (its too pest sensitive) so pesticide and fertiliser use is standard, although again poorly recorded in terms of detail between studies.

Comment: line 590 - This figure is not presented in the main body of the manuscript!

Response: Its now mentioned

Comment: line 591 - Delete "and" after abundance. Replace by comma.

Response: Changed.

Comment: line 598 - This is just a repetition of the title of the figure. Please revise for more original sentence structure or delete.

Response: This is true but journal format means I cant include parenthesis here giving crucial info like the use of CI and the abbreviation for r. However, I have changed the title so its not repetitive to 'Forest plots for the relationship between oilseed rape yield and pollinator community structure'.

Comment: line 602- Insert comma after "insect abundance".

Response: Changed.

Comment: line 609 - It would be informative to include year of study in the table. It would also be helpful to have a map showing where all of the studies were conducted, to have an idea of its international scope. Although conclusions are drawn on the "global" ramifications of this meta-analysis (line 71), the studies are actually rather Euro-centric in their distribution and there's virtually no representation from the central hemisphere.

Response: We now acknowledge in the methods section the geographical limitations of the study set (specifically Northern Hemisphere and Europe based). We do not feel a map is required, given the ultimate number of studies and the fact that table 2 clearly states where each study is from. As year is not included in the analysis we don't see this as relevant information, combined with the fact that the study date is given in the published reference of supplementary methods. Note we downplay in response to your earlier comments the 'global' assertions we made in the abstract now (L71 comment). In the conclusion we also now have the line '*These results are, however, based on a data set biased to the Northern Hemisphere, and in particular Europe, and as such may not necessarily be generalisable to other regions where the crop is grown or interacts with functionally different pollinator communities.*'

Comment: line 613 - Delete "include".

Response: Changed.

Comment: line 624- Change to "... are based on observations of wild pollinator communities under..."

Response: Changed.

Table 3

Comment: effect trait 1 - change "characterises" to "characterized"; change "affect" to "affects"

Response: Changed as suggested.

Comment: effect traits 2,3-4,5 and 6 need to be formatted so that the phrase "but augmented and additional with unpublished data" is not superscript. It also needs to be changed to simply "but augmented with unpublished data". Insert a space between the closed parenthesis and the word "based".

Response: Changed as suggested and in response to below comments.

Comment: effect trait 7 - I had to reformat this whole paragraph. Please review and see where I made changes.

“Hairiness affects pollen grain deposition on stigmas 51 and, in bees, is used to detect electromagnetic fields emitted by flowers as pollination cues 66. For each species, body parts that contact oilseed stigmas (head, thorax, sternum, abdomen underside, femora, tibiae and meta-tarsus (legs assessed separately)) were scored as: 1) coarse setae or extremely short hairs; 2) short (c. basal tibiae 1 diameter) but dense hairs (>50 mm²); 3) long (>basal tibiae 1 diameter) dense (>50 mm²) hairs. This score was summed and given as a percentage of the maximum score of 24.

Response: Changed as suggested.

Comment: N.B. What is meant by "1 diameter"? Is a unit measure missing here?

Response: changed to read '(c. length of the diameter of the basal tibiae)'

Comment: effect trait 8- change "Tongue" to "The length of the tongue" or "The morphology of the tongue". Also change to "A separate category is listed for insects with chewing mouthparts".

Response: Changed as suggested.

Comment: effect traits 9 - 13 : please note the changes that I've made to the following paragraph:

The presence of setae specifically used to collect pollen, listed by Michener 52 as the basitarsal scopa (trait 9), femoral corbicula (trait 10), strict tibial corbicula (trait 11), propodeal corbicula (trait 12) or abdominal corbicula (trait 13). Note these structures are associated with bees, however, their absence will affect the pollen carrying capacity and thus likelihood of pollen stigmal transfer of other species (e.g. for flies).

Response: Changed as suggested.

Comment: effect trait 14: please note the changes that I've made to this section:

Pollen carried only in the crop and, as such, not available for pollination 52. As above, these structures are associated with bees, however, their absence may affect the likelihood of pollen contacting plant stigmas for other pollinating groups.

Response: Changed as suggested.

Comment: effect trait 15: Please note the changes in the following paragraph:

Pollen in corbicula storage structures may be either dry or moistened. Moistened pollen is less freely available for deposition onto plant stigmas 52. As above, these structures are

associated with bees, however, their absence may affect the likelihood of pollen contacting plant stigmas for other pollinating groups.

Response: Changed as suggested.

Comment: Fig. 1 - The three panels need to be aligned on the lower horizontal axis. Each panel appears to have been narrowed individually so that the dots representing the mean values are more oval than circular. Ideally, the points would be circular. This whole figure was not mentioned in the text. An x-axis title would be helpful.

Response: Figures are aligned and narrowing removed so symbols are circular. X axis added. Figure is now mentioned in text.

Comment: Fig. 2 - I really appreciate the formatting of this figure, as it's quite easy to interpret. However, It would be helpful to be presented with scatter plots or partial effect graphs for each predictor variable to get a sense of the distribution of the data or the residuals (https://www.researchgate.net/publication/241653683_An_Effect_Size_for_Regression_Predictors_in_Meta-Analysis). Perhaps this could be provided as Supplemental Material in addition to the funnel plots. It would be used to see whether the significant relationships are not being driven by a subset of studies. Moreover, the values beginning with $\mu = \#$ [#, #] need to be aligned right.

Response: We now include scatter plots for the raw data for each study and predictor variable of yield. These are given in supplementary fig. 1.. Note these same graphs are also used for the interpretation of the general linear mixed modelling we have undertaken in response to your comments above (the raw plots are most useful as the random effects mean that each study has a unique intercept and so sticking them all into one graph ends up being a bit cluttered).

References

- Gagic V., Bartomeus I., Jonsson T., Taylor A., Winqvist C., Fischer C., Slade E.M., Steffan-Dewenter I., Emmerson M., Potts S.G., Tscharrntke T., Weisser W. & Bommarco R. (2015) Functional identity and diversity of animals predict ecosystem functioning better than species-based indices. *Proceedings of the Royal Society B: Biological Sciences*, **282**.
- Garibaldi L.A., Steffan-Dewenter I., Kremen C., Morales J.M., Bommarco R., Cunningham S.A., Carvalheiro L.G., Chacoff N.P., Dudenhoffer J.H., Greenleaf S.S., Holzschuh A., Isaacs R., Krewenka K., Mandelik Y., Mayfield M.M., Morandin L.A., Potts S.G., Ricketts T.H., Szentgyorgyi H., Viana B.F., Westphal C., Winfree R. & Klein A.M. (2011) Stability of pollination services decreases with isolation from natural areas despite honey bee visits. *Ecology Letters*, **14**, 1062-1072.
- Hoehn P., Tscharrntke T., Tylianakis J.M. & Steffan-Dewenter I. (2008) Functional group diversity of bee pollinators increases crop yield. *P. Roy. Soc B-Biol. Sci.*, **275**, 2283-2291.
- Jamil T., Ozinga W.A., Kleyer M. & ter Braak C.J.F. (2013) Selecting traits that explain species–environment relationships: a generalized linear mixed model approach. *Journal of Vegetation Science*, **24**, 988-1000.
- Mason N.W.H., Mouillot D., Lee W.G. & Wilson J.B. (2005) Functional richness, functional evenness and functional divergence: the primary components of functional diversity. *Oikos*, **111**, 112-118.
- Petchey O. & Gaston K.J. (2006) Functional diversity: back to basics and looking forward. *Ecology Letters*, **9**, 741-758.

Sonnier G., Shipley B. & Navas M.-L. (2010) Quantifying relationships between traits and explicitly measured gradients of stress and disturbance in early successional plant communities. *Journal of Vegetation Science*, **21**, 1014-1024.

Reviewers' Comments:

Reviewer #2:

Remarks to the Author:

I consider authors have made a hard and good job addressing all comments from the three referees, even they did not include all suggestions; they have sufficiently argued their responses.

In this sense, I think author have solved all the stated questions, and also specific and methodological details. However, I suggest a couple of general but important comments they should address to be suitable in Nature Communications.

(1) I understand the doubts and comments from the others referees on the methods and Results sections. I also understand why authors have now included more information in these two sections. However, from my point of view, Results section is currently presented as too much 'statistical'. I mean, I would use a more simpler language, helping the reader to easier flowing through the text and better catch the key messages of results, but from a ecological perspective. I leave to the Editor the decision on this potential change. It is not a specific comment, instead a suggestion for a small 'cleanning' of the section to make it more fluent and friendly.

(2) I'd highlighted the geographical constraints of the study (little biased to Northern Hemisphere and Europe). Even authors have now included mentions of these limitations, I consider they may bring some discussion on the potential implications or consequences in other regions where canola is produced. At the same time, I suggest authors also including some mention to other systems (pollinator-plants). If this is not done, the paper will still remain restricted to specific regions and cases, while it is providing information, data and results enough robust to discuss implications from a more broader perspective (both in terms of regions and species). I consider this is a must bearing in mind the paper will be published in a high impact and broad scope journal as Nature Communications is.

(3) Only a minor correction in lines 124-128. I would remove the specifications on field studies and mesocosms, as they are better explained in the next section. Thus, I suggest move these details to paragraph in lines 141-153. They are more results than goals, aren't they?

Reviewer #3:

Remarks to the Author:

I appreciate all of the revisions that the authors have implemented since the previous version of the manuscript. Bravo! I do, however, still have some concerns about the conclusions drawn from the analyses. My main concern pertains to the data distribution of the variable FDiv. By examining the supplemental figures, it appears as if it more closely models the presence or absence of pollinators at field sites as opposed to the complementarity of functional traits. I have included in this review an Rmarkdown document to test this, drawing on data included with the file "Supplemental Dataset". Please consider this assessment and respond to it by modifying the analysis accordingly should the article carry through to an additional round of reviews.

Additional Notes:

line 99 - Why is "originating front he plant community literature" separated by ,

line 115 - "except Antarctica" not in ,

line 127 - Awkward phrasing - Please change to "The field studies were predominantly from Europe,

but some were from the USA and China."

line 141 - Meta-analyses is still pluralized. The study itself comprises one meta-analysis, so I'm unsure why the authors are implying that many were conducted. Perhaps I am misunderstanding something.

line 141 - the phrasing should be "The data set used in the meta-analysis was based...". "Was" should conjugate with "data set" not "analyses".

line 142 - This sentence is hard to understand. How can a "correlation" be a "unit" in a "meta-analysis"? Please clarify and rephrase.

line 144 - This sentence also makes little sense and is not very readable. Consider removing "as an overview of the source data from which the correlations were derived". State simply that "The 23 studies contained...".

line 146 - Change to "These taxonomic units included species level (N=36) and genus level (N=19) classifications, as well as functional groups...". The term "generic" is missing a complement as currently phrased.

line 148 - Include comma after "conditions".

line 180 - Replace "1" with "one".

line 187 - What was the model structure ? were there 16 observations and 15 + fixed effects ?

line 259 - Reconsider use of "niche"

line 270 - Remove comas around "and thus more biologically realistic". I suspect that this statement is false. There is actually substantially more variance associated with the variable effect group richness as when compared with that of functional divergence. This suggests that effect group richness approximates a more gradual gradient in trait distributions, which would in turn be more "biologically realistic". FDiv seems to be rather discrete as calculated. Please see Rmarkdown for a further elaboration of this idea.

line 281 - Remove coma after "contact"

line 285 - This is an interesting observation but what evidence is there to suggest that this might be the case for oil seed rape?

line 292 - Remove coma after "unmeasured"

line 157 and 158 - change 2 to two

Rev#1

No comments provided.

Rev#2

Comment: I consider authors have made a hard and good job addressing all comments from the three referees, even they did not include all suggestions; they have sufficiently argued their responses. In this sense, I think author have solved all the stated questions, and also specific and methodological details. However, I suggest a couple of general but important comments they should address to be suitable in Nature Communications.

Response: We are happy we have addressed these issues

Comment: (1) I understand the doubts and comments from the others referees on the methods and Results sections. I also understand why authors have now included more information in these two sections. However, from my point of view, Results section is currently presented as too much 'statistical'. I mean, I would use a more simpler language, helping the reader to easier flowing through the text and better catch the key messages of results, but from a ecological perspective. I leave to the Editor the decision on this potential change. It is not a specific comment, instead a suggestion for a small 'cleaning' of the section to make it more fluent and friendly.

Response: We feel that the language is clear. For example there are always clear statements of the relationship between a community metric and yield, e.g. '*For both mesocosm and field studies positive correlations were identified with yield (L157-159)*'. This is the fundamental result of interest and we are not sure how to make it much clearer. Partly we feel the issue is the need to present the test statistics (which is unavoidable) which does make the text denser. That said we feel that the Forest plots for the meta-analysis, when presented in context with the paper, make the results clear as this graphical presentation is intuitively obvious as to where positive effects of community metrics on yield were identified. We are also somewhat constrained in the language, partly because of the need to present statistically robust statements of the effects, but also in the fact that some of this detail was requested in previous iterations of the paper by another referee. If the editor feels that this language

needs to be significantly changed to favour your opinion as opposed to that of the other referees comments we would be happy to do it, but we personally believe that the current framing in combination is the forest plot is appropriate.

Comment: I'd highlighted the geographical constraints to the study (little biased to Northern Hemisphere and Europe). Even authors have now included mentions of these limitations, I consider they may bring some **discussion on the potential implications or consequences in other regions where canola is produced**. At the same time, I suggest authors also including some mention to other systems (pollinator-plants). If this is not done, the paper will still remain restricted to specific regions and cases, while it is providing information, data and results enough robust to discuss implications from a more broader perspective (both in terms of regions and species). I consider this is a must bearing in mind the paper will be published in a high impact and broad scope journal as Nature Communications is.

Response: As you have noted we now make it very clear that while this contains studies from across the world it is not comprehensive. We state clearly the geographical range both directly in the text and in tables describing the origins of individual studies. However, we now include the below discussion point which highlights the relevance of the findings to other regions that may be growing this crop, but not represented in the analysis.

'While this hypothesis was supported by a correlation between functional divergence in effect traits and yield, these findings are based on a data set biased to the Northern Hemisphere and Europe in particular. As such, these results may not necessarily be generalizable to other regions, particularly if those regions are characterised by functionally different pollinator communities interacting with the same crop. However, the effect of complementarity on yield, as predicted by community functional divergence, may be expected to have relevance to other regions where oilseed rape is grown. Even where a fauna taxonomically distinct to those considered here forages on this crop there would likely be equivalent variation in the considered insect pollinator effects traits. An increase in functional divergence as explained by these effect traits could similarly be expected to have a positive impacts on oilseed rape yield. '

We also include the below lines to highlight the relevance of the study to other crop types.

'While this study focuses on oilseed rape, it has important implications for the role of insect pollination in general. However, crops with different breeding types or morphologically distinct flowers may have different dependencies on insect pollinators, with distinct effect traits to those considered here potentially having greater significance in terms of their impacts on yield. Further research is required to refine these relationships to maximise the potential for targeted management to support agricultural production.'

Comment: Only a minor correction in lines 124-128. I would remove the specifications on field studies and mesocosms, as they are better explained in the next section. Thus, I suggest move these details to paragraph in lines 141-153. They are more results than goals, aren't they?
Response: As requested this section has now been moved to the opening paragraph of the results.

Rev#3

Comment: I appreciate all of the revisions that the authors have implemented since the previous version of the manuscript. Bravo! I do, however, still have some concerns about the

conclusions drawn from the analyses. My main concern pertains to the data distribution of the variable FDiv. By examining the supplemental figures, it appears as if it more closely models the presence or absence of pollinators at field sites as opposed to the complementarity of functional traits. I have included in this review an Rmarkdown document to test this, drawing on data included with the file "Supplemental Dataset". Please consider this assessment and respond to it by modifying the analysis accordingly should the article carry through to an additional round of reviews.

Response: This is a fair point. In cases of studies that included control plots (where pollinators were excluded) the use of raw FDiv resulted in a covariate that was qualitatively similar to a binary covariate describing plots as with or without bees. We address this issue in two complementary ways for functional divergence, each of which provide the same answer (e.g. a positive effect of functional divergence on yield). Approach 1) We derived correlations between FDiv (unscaled) and yield, but excluded all control plots (i.e. the zero pollinator abundance plots). This is a robust way of dealing with this issue as it only includes relationships between yield and functional diversity where bees were present, however, this does exclude the important comparison to the no pollinator control which is typically used in the determination of pollinator effects on yield; 2) To address this issue of the loss of the no-pollinator controls we produce a scaled functional diversity metric. For each study all FDiv values > 0 were corrected to be FDiv - lowest non-zero FDiv value for the study. This keeps zero pollinator controls but rescales FDiv to have meaningful variation for assessing a correlation with yield. Both results are presented in the paper and we feel in combination (and individually) address your valid concern. All correlations for individual studies functional divergence and yield are presented in the supplementary material, including the original FDiv index (to highlight the problem of using it in its raw form although this is not now used in the main paper), FDiv scaled and FDiv without control plots.

Comment: line 99 - Why is "originating front he plant community literature" separated by ,

Response: Comma removed.

Comment: line 115 - "except Antarctica" not in ,

Response: I am not entirely clear what change you want here. I am assuming you are wanting me to change 'on' into 'in' to read '*This crop is grown in all continents, except Antarctica, and is one of the principal crops used in the production of edible oils and biodiesel.*'

Comment: line 127 - Awkward phrasing - Please change to "The field studies were predominantly from Europe, but some were from the USA and China."

Response: Changed as suggested.

Comment: line 141 - Meta-analyses is still pluralized. The study itself comprises one meta-analysis, so I'm unsure why the authors are implying that many were conducted. Perhaps I am misunderstanding something.

Response: This is incorrect, the study undertakes multiple meta-analyses, one for each of the community covariates (such as SR, FDiv etc – these effectively represent response variables).

Comment: line 141 - the phrasing should be "The data set used in the meta-analysis was based...". "Was" should conjugate with "data set" not "analyses".

Response: Changed as suggested, though note we use analyses as there were more than one undertaken.

Comment: line 142 - This sentence is hard to understand. How can a "correlation" be a "unit" in a "meta-analysis"? Please clarify and rephrase.

Response: Sentence changed to read *'From each study correlations between oilseed rape yield and measures of pollinator community structure were derived. The relative strength and direction of this correlations were then assessed within the meta-analyses.'*

Comment: line 144 - This sentence also makes little sense and is not very readable. Consider removing "as an overview of the source data from which the correlations were derived". State simply that "The 23 studies contained...".

Response: Changed as suggested.

Comment: line 146 - Change to "These taxonomic units included species level (N=36) and genus level (N=19) classifications, as well as functional groups...". The term "generic" is missing a complement as currently phrased.

Response: Changed to suggested text.

Comment: line 148 - Include comma after "conditions".

Response: Added

Comment: line 180 - Replace "1" with "one".

Response: Changed

Comment: line 187 - What was the model structure ? were there 16 observations and 15 + fixed effects ?

Response: We now include the clarifying line in the results section *'We used general linear mixed models to test for correlations between oilseed rape yield (as a response) and linear combinations of the 15 effect traits described by their community weighted means (CWM) as explanatory variables'*. The real detail of the approach is given in the stats section but this clearly outlines exactly what we did in the context of the results.

Comment: line 259 - Reconsider use of "niche"

Response: line changed to read *'Moving beyond simple species richness, the number of clusters of species interacting with the crop in biologically similar ways provides as indication of how many functional distinct groups of species are pollinating a crop'*.

Comment: line 270 - Remove comas around "and thus more biologically realistic". I suspect that this statement is false. There is actually substantially more variance associated with the variable effect group richness as when compared with that of functional divergence. This suggests that effect group richness approximates a more gradual gradient in trait distributions, which would in turn be more "biologically realistic". FDiv seems to be rather discrete as calculated. Please see Rmarkdown for a further elaboration of this idea.

Response: Firstly we have removed the comas as suggested. In response to your above comments the FDiv index has been reanalysed using two complementary methods that address the issues relating to the lack of variation in this index (i.e. the discrete nature of FDiv as originally used is no longer an issue). We now feel that this statement is therefore justified. We would of course still be happy to remove it if it was felt necessary.

Comment: line 281 - Remove coma after "contact"

Response: removed

Comment: line 285 - This is an interesting observation but what evidence is there to suggest that this might be the case for oil seed rape?

Response: We now include the line '*However, at present there is no direct evidence that this may be occurring in oilseed rape.*'

Comment: line 292 - Remove coma after "unmeasured"

Response: removed.

Comment: line 157 and 158 - change 2 to two

Response: Changed.

Reviewers' Comments:

Reviewer #3:

Remarks to the Author:

Reviewer 3

Overall, I appreciate all the work the authors put in to addressing the comments made in the previous round of revisions. This especially pertains to their transformation of FDiv to account for its binary distribution when control plots are included in the analysis. I think their new approach and treatment of the variable is valid. Overall, the analysis is much stronger but I still had some issues with the phrasing of the text. Rereading the manuscript has allowed me to pick up on errors and typos that I hadn't seen before. Please consider the comments below prior to making a new submission.

Line 61 – the authors emphasize that they have conducted multiple meta-analyses but refer to a single meta-analysis here. Please revise for consistency.

Line 76 - contradicts the opening line of the abstract. It suggests that abundance and richness are well understood in terms of their contribution to crop pollination, as opposed to functional differences, but line 59 suggests that "overall abundance" and "species richness" are also poorly understood. Please rectify this by modifying line 59 to be consistent with line 76.

Line 116 – My apologies for not being clear in my recommendation for this line in the previous round of reviews. In fact, I was suggesting that you not isolate "except Antarctica" between commas. This is a stylistic choice and I leave it to the journal editor to decide whether the commas are warranted.

Line 119 – add "pollination on yield in oilseed rape".

Line 127 – the punctuation here is strange. The semi-colon makes an implicit link between the sentence starting on line 125 with the second point being tested, as denoted by "(2)". This link is nonsensical, as the line starting on 125 is actually an elaboration of the first item in this list. Here's what I would suggest: "We infer... with oilseed rape and yield. We also test the extent to which (2) pollination is determined by...mass ratio hypothesis".

Line 142 – insert comma after "study". I would also suggest a more straightforward and active phrasing, namely: "From each study, we correlated oilseed rape yield and measures of pollinator community structure".

Line 143 – change "this correlations" to "these correlations" . Suggested rephrasing: "We then assessed the relative strength and direction of these correlations for each meta-analysis".

Line 147 – 9 – the abbreviation for species is inconsistently formatted. There seems to be many species of *Pieris* so you likely mean "*Pieris* spp.". I don't see why "spp." should be italicized after "Calypterate". Please also check that Calypterate is not a misspelling of calyprate. Finally, as calyprate is an adjective it seems more appropriate to say "calyprate flies".

Line 151 – suggest removing the comma between "communities" and "with"

Lines 158, 161, 164, 181, 184, 219, 220, 234 – Should be "Cook's distance" not "Cooks distance"

Line 159 – Should be "positive correlations were identified between yield and abundance".

Line 171 – Makes little sense to "quantify" a test. I think you mean : "We quantified the role played by complementarity in species traits by testing the relationship between functional divergence and effect group richness on oilseed rape yield".

Line 179 – Change to : "This was true when using either... or an unscaled"

Line 244 – Change "insect" to "insects"

Line 246- insert comma before "this"

Line 267 – change "as" to "an"

Line 268 – change "functional" to "functionally"

Line 282- recommend including a citation here to Martins, Gonzalez, Lechowicz 2015 AEE as evidence of a complementary effect in action in another crop pollination system. Could also be mentioned on line 300 as the article also showed some complementarity between *Andrena* and *Bombus*.

Line 289 – change "interacts" to "interact"

Line 299- there's no real need to specify "the genus *Andrena* spp."; you can just omit "spp.". I think the same is true of "*Bombus*" in this sentence and avoids the awkward double period after the second "spp."

Line 303 – ensure that the journal prefers British as opposed to American spelling of the word "emphasize".

Line 305 – This whole sentence needs to be rewritten. For instance, you repeat yourself by saying "...although from an applied pollinator standpoint community functional diversity alone...has greater direct relevance from a management perspective". Consider changing the sentence to "Identifying an underpinning and independent influence of insect phylogeny on pollination may be more pertinent to the study of evolution as opposed to crop management". At the very least, please attempt to simplify the sentence structure here.

Line 314 – impact should conjugate with role not traits, so please replace with "impacts". You can also rephrase this to "the complementary role of specific effect traits impacts pollination success"

Line 317 – consider removing "correlated with stigma deposition rates". Otherwise, the sentence is too wordy and this correlation is already suggested through the usage of "key effect traits". Actually, "communities dominated by species with key effect traits" doesn't really reflect the CWM calculation. You tabulated these key traits for both common and uncommon species. It's not like the abundant species had more of these key traits than the rare ones. It's more like you weighted the mean of these traits for each community in terms of the most abundant species. I think you mean to say, "the effect traits of dominant species did not correlate positively with oilseed rape yield".

Line 320 – If you follow my recommendation for Line 317, then you can simply state "As such, we found no direct evidence in support of the mass ratio hypothesis". Please insert "of" either way.

Line 323 and 324 – I think you're conflating an ecological and a mathematical argument here. Yes, species abundances at different sites are a product of a variety of factors, but this doesn't explain why the mean and total abundance may or may not be correlated. I think it would mean rather that the abundant species tended to have similar traits across studies. Please clarify.

Line 346- the phrasing here doesn't make sense. I suspect there are typos related to the words "where", "her" and the conjugation of "forages" but I can't make sense of what you're trying to say to make any real recommendation.

Line 350- There seem to be many typos in this recent round of revisions. "Impacts" should be "impact".

Line 352- change "by the location of" to "by placing".

Line 353 – change to "increase the functional divergence of the overall community"

Line 355 – management is typically used as an adjective with this phrasing so consider "Such management tactics"

Line 357 – the comma needs to be placed after "or" . Also, replace "though" with "through".

Line 359 – "could be respectively be used" doesn't make sense. Change to "could be used to promote... species (e.g. some *Bombus* spp.), respectively".

Line 389 - change "inclusions" to "inclusion". Also, should "meta-analysis" be pluralized here for consistency?

Line 401 – why is 18 italicized ? Also, it should be "This resulted in..."

Line 411 – change "we do not" to "we did not". Also insert a comma after "studies".

Line 411 – Change to "While landscape setting" as opposed to "While landscape"

Line 413 – change to "rather than from where the pollinators originated"

Line 414 – insert comma after "experiments"

Line 415- Change to "The dependency of the ... and the attractiveness of the crop to pollinators have been shown to...". Also consider changing "have been shown to be affected by" to "are affected by".

Line 429 – change the comma after "23 studies" to a semi-colon. Also, "both exceptions occurred under field conditions".

Lines 442, 460 – change "1" to "one".

Line 450 – Again, do you mean "*Delia* spp."?

Line 452- insert comma after)

Line 459 – remove “applied”

Line 470 – delete “where” after the comma,

Line 471- change to “with mean trait values”

Line 472 – delete “were derived”

Line 478 – “An” shouldn’t be capitalized

Lines 482 – 483 – please move this sentence to the end of the paragraph

Lines 485, 594 – is this really the recommended format for a in-text citation ? Shouldn’t you just include the superscript footnote? Please review the author guidelines.

Line 491 – change to “the spread of the niche space”

Line 493 – change “Wards algorithm” to “Ward’s method”, insert comma after groups, also it should be “To define the effect groups”.

Line 514 – change to “For studies that included control plots, FDiv was quantitatively similar to a binary covariate describing plots with or without bees. We applied two separate approaches to address this issue, the first being to rescale our measure of functional diversity while retaining a comparison with control plots lacking pollinators. For each study in which FDiv values were greater than zero, values were corrected to FDiv minus the lowest non-zero FDiv value for that study. The second approach was to derive correlations between yield and FDiv after having excluded all control plots.”

Line 535 – remove comma after napus

Line 536 – include comma after “rates”

Line 798 – I think you mean “Pearson’s” not “Parson’s”

Line 815 – change to “... for measures when excluding pollinator control plots”. I don’t think you mean “excluding no”, suggesting “without excluding”.

Line 819 – consider removing “used in meta-analysis”. If you keep it, consider inserting “the meta-analyses”.

Response to reviewer comments

All reviewers' comments are highlighted in yellow in the manuscript.

Reviewer #3 (Remarks to the Author):

Reviewer 3

Comment: Overall, I appreciate all the work the authors put in to addressing the comments made in the previous round of revisions. This especially pertains to their transformation of FDiv to account for its binary distribution when control plots are included in the analysis. I think their new approach and treatment of the variable is valid. Overall, the analysis is much stronger but I still had some issues with the phrasing of the text. Rereading the manuscript has allowed me to pick up on errors and typos that I hadn't seen before. Please consider the comments below prior to making a new submission. **Response:** We thank you for your very detailed comments and have implemented all your comments including all suggestions for text rephrasing as outlined in our response to you below. All changes are highlighted in the manuscript.

Comment: Line 61 – the authors emphasize that they have conducted multiple meta-analyses but refer to a single meta-analysis here. Please revise for consistency. **Response:** Changed

Comment: Line 76 - contradicts the opening line of the abstract. It suggests that abundance and richness are well understood in terms of their contribution to crop pollination, as opposed to functional differences, but line 59 suggests that “overall abundance” and “species richness” are also poorly understood. Please rectify this by modifying line 59 to be consistent with line 76. **Response:** Sentence in L 59 of abstract now reads ‘How insects promote crop pollination remains poorly understood in terms of the contribution of functional trait differences between species’

Comment: Line 116 – My apologies for not being clear in my recommendation for this line in the previous round of reviews. In fact, I was suggesting that you not isolate “except Antarctica” between commas. This is a stylistic choice and I leave it to the journal editor to decide whether the commas are warranted. **Response:** Changed as suggested

Comment: Line 119 – add “pollination on yield in oilseed rape”. **Response:** Changed

Comment: Line 127 – the punctuation here is strange. The semi-colon makes an implicit link between the sentence starting on line 125 with the second point being tested, as denoted by “(2)”. This link is nonsensical, as the line starting on 125 is actually an elaboration of the first item in this list. Here's what I would suggest: “We infer... with oilseed rape and yield. We also test the extent to which (2) pollination is determined by...mass ratio hypothesis”.

Response: I disagree with the 'oilseed rape and yield' suggestion as its oilseed rape yield specifically we are correlating these metrics with. However, we change the sentences as otherwise indicated.

Comment: Line 142 – insert comma after “study”. I would also suggest a more straightforward and active phrasing, namely: “From each study, we correlated oilseed rape yield and measures of pollinator community structure”. **Response:** Changed to alternate sentence

Comment: Line 143 – change “this correlations” to “these correlations” . Suggested rephrasing: “We then assessed the relative strength and direction of these correlations for each meta-analysis”. **Response:** Changed to alternate sentence

Comment: Line 147 – 9 – the abbreviation for species is inconsistently formatted. There seems to be many species of *Pieris* so you likely mean “*Pieris* spp.”. I don’t see why “spp.” should be italicized after “Calypterate”. Please also check that Calypterate is not a misspelling of calyprate. Finally, as calyprate is an adjective it seems more appropriate to say “calyprate flies”. **Response:** All changed as suggested

Comment: Line 151 – suggest removing the comma between “communities” and “with”
Response: removed

Comment: Lines 158, 161, 164, 181, 184, 219, 220, 234 – Should be “Cook’s distance” not “Cooks distance” **Response:** Changed, and updated in supplementary material

Comment: Line 159 – Should be “positive correlations were identified between yield and abundance”. **Response:** Changed

Comment: Line 171 – Makes little sense to “quantify” a test. I think you mean : “We quantified the role played by complementarity in species traits by testing the relationship between functional divergence and effect group richness on oilseed rape yield”. **Response:** Changed as suggested

Comment: Line 179 – Change to : “This was true when using either... or an unscaled”
Response: changed

Comment: Line 244 – Change “insect” to “insects” **Response:** changed

Comment: Line 246- insert comma before “this” **Response:** added

Comment: Line 267 – change “as” to “an” **Response:** changed

Comment: Line 268 – change “functional” to “functionally” **Response:** Changed

Comment: Line 282- recommend including a citation here to Martins, Gonzalez, Lechowicz 2015 AEE as evidence of a complementary effect in action in another crop pollination system. Could also be mentioned on line 300 as the article also showed some complementarity between *Andrena* and *Bombus*. **Response:** Added in both places

Comment: Line 289 – change “interacts” to “interact” **Response:** changed

Comment: Line 299- there’s no real need to specify “the genus *Andrena* spp.”; you can just omit “spp.”. I think the same is true of “*Bombus*” in this sentence and avoids the awkward double period after the second “spp.” **Response:** Removed as suggested

Comment: Line 303 – ensure that the journal prefers British as opposed to American spelling of the word “emphasize”. **Response:** I checked its English spelling.

Comment: Line 305 – This whole sentence needs to be rewritten. For instance, you repeat yourself by saying “...although from an applied pollinator standpoint community functional diversity alone...has greater direct relevance from a management perspective”. Consider changing the sentence to “Identifying an underpinning and independent influence of insect phylogeny on pollination may be more pertinent to the study of evolution as opposed to crop management”. At the very least, please attempt to simplify the sentence structure here. **Response:** Changed to your suggested sentence.

Comment: Line 314 – impact should conjugate with role not traits, so please replace with “impacts”. You can also rephrase this to “the complementary role of specific effect traits impacts pollination success” **Response:** Change to the alternate sentence.

Comment: Line 317 – consider removing “correlated with stigma deposition rates”. Otherwise, the sentence is too wordy and this correlation is already suggested through the usage of “key effect traits”. Actually, “communities dominated by species with key effect traits” doesn’t really reflect the CWM calculation. You tabulated these key traits for both common and uncommon species. It’s not like the abundant species had more of these key

traits than the rare ones. It's more like you weighted the mean of these traits for each community in terms of the most abundant species. I think you mean to say, "the effect traits of dominant species did not correlate positively with oilseed rape yield". **Response:** changed as suggested

Comment: Line 320 – If you follow my recommendation for Line 317, then you can simply state "As such, we found no direct evidence in support of the mass ratio hypothesis". Please insert "of" either way. **Response:** changed as suggested

Comment: Line 323 and 324 – I think you're conflating an ecological and a mathematical argument here. Yes, species abundances at different sites are a product of a variety of factors, but this doesn't explain why the mean and total abundance may or may not be correlated. I think it would mean rather that the abundant species tended to have similar traits across studies. Please clarify. **Response:** We have deleted this sentence

Comment: Line 346- the phrasing here doesn't make sense. I suspect there are typos related to the words "where", "her" and the conjugation of "forages" but I can't make sense of what you're trying to say to make any real recommendation. **Response:** Changed to read '*Even where a fauna is taxonomically distinct to that considered here these novel communities would likely still show similar levels of variation in the effects traits we have considered.*'

Comment: Line 350- There seem to be many typos in this recent round of revisions. "Impacts" should be "impact". **Response:** changed

Comment: Line 352- change "by the location of" to "by placing". **Response:** changed

Comment: Line 353 – change to "increase the functional divergence of the overall community" **Response:** changed

Comment: Line 355 – management is typically used as an adjective with this phrasing so consider "Such management tactics" **Response:** changed

Comment: Line 357 – the comma needs to be placed after "or" . Also, replace "though" with "through". **Response:** changed

Comment: Line 359 – "could be respectively be used" doesn't make sense. Change to "could be used to promote... species (e.g. some *Bombus* spp.), respectively". **Response:** Changed

Comment: Line 389 - change “inclusions” to “inclusion”. Also, should “meta-analysis” be pluralized here for consistency? **Response:** Changed

Comment: Line 401 – why is 18 italicized ? Also, it should be “This resulted in...” **Response:** changed

Comment: Line 411 – change “we do not” to “we did not”. Also insert a comma after “studies”. **Response:** changed

Comment: Line 411 – Change to “While landscape setting” as opposed to “While landscape”
Response: changed

Comment: Line 413 – change to “rather than from where the pollinators originated”
Response: changed

Comment: Line 414 – insert comma after “experiments” **Response:** Added

Comment: Line 415- Change to “The dependency of the ... and the attractiveness of the crop to pollinators have been shown to...”. Also consider changing “have been shown to be affected by” to “are affected by”. **Response:** Changed

Comment: Line 429 – change the comma after “23 studies” to a semi-colon. Also, “both exceptions occurred under field conditions”. **Response:** changed.

Comment: Lines 442, 460 – change “1” to “one”. **Response:** changed

Comment: Line 450 – Again, do you mean “*Delia* spp.”? **Response:** changed

Comment: Line 452- insert comma after) **Response:** added

Comment: Line 459 – remove “applied” **Response:** deleted

Comment: Line 470 – delete “where” after the comma, **Response:** deleted

Comment: Line 471- change to “with mean trait values” **Response:** changed

Comment: Line 472 – delete “were derived” **Response:** deleted.

Comment: Line 478 – “An” shouldn’t be capitalized **Response:** changed.

Comment: Lines 482 – 483 – please move this sentence to the end of the paragraph
Response: moved to the paragraph end.

Comment: Lines 485, 594 – is this really the recommended format for a in-text citation ?
Shouldn’t you just include the superscript footnote? Please review the author guidelines.
Response: Changed to superfix format only.

Comment: Line 491 – change to “the spread of the niche space” **Response:** Changed.

Comment: Line 493 – change “Wards algorithm” to “Ward’s method”, insert comma after
groups, also it should be “To define the effect groups”. **Response:** changed as suggested.

Comment: Line 514 – change to “For studies that included control plots, FDiv was
quantitatively similar to a binary covariate describing plots with or without bees. We applied
two separate approaches to address this issue, the first being to rescale our measure of
functional diversity while retaining a comparison with control plots lacking pollinators. For
each study in which FDiv values were greater than zero, values were corrected to FDiv
minus the lowest non-zero FDiv value for that study. The second approach was to derive
correlations between yield and FDiv after having excluded all control plots.” **Response:**
Changed to the suggested text

Comment: Line 535 – remove comma after napus **Response:** removed

Comment: Line 536 – include comma after “rates” **Response:** added

Comment: Line 798 – I think you mean “Pearson’s” not “Parson’s” **Response:** Changed

Comment: Line 815 – change to “... for measures when excluding pollinator control plots”. I don’t think you mean “excluding no”, suggesting “without excluding”. **Response:** Changed as suggested.

Comment: Line 819 – consider removing “used in meta-analysis”. If you keep it, consider inserting “the meta-analyses”. **Response:** Text removed